

# Volcanic impact on the climate – the stratospheric aerosol load in the period 2006 – 2015

Johan Friberg[1], Bengt G. Martinsson[1], Sandra M. Andersson[1,2], Oscar S. Sandvik[1]

[1]Division of Nuclear Physics, Lund University, Lund, 22100, Sweden
[2] Now at the Swedish Meteorological and Hydrological Institute, Norrköping, Sweden

*Correspondence to*: Johan Friberg (johan.friberg@nuclear.lu.se)

**Abstract.** We present a study on the stratospheric aerosol load during 2006-2015, discuss the influence from volcanism and other sources, and reconstruct an AOD data-set in a resolution of 1° latitudinally and 8 days timewise. Our attempt is to include the "entire" stratosphere, from the tropopause to the almost particle free altitudes of the mid-stratosphere. A dynamic tropopause of 1.5 PVU was found to be the best suited one, enclosing almost all of the volcanic signals in the CALIOP data-

set. New methods were developed to handle bias in the data. The data were successfully cleaned from polar stratospheric clouds using a temperature threshold of 195 K. Furthermore, a method was developed to correct data when the CALIPSO laser beam was strongly attenuated by volcanic aerosol, preventing a negative bias in the AOD data-set. Tropospheric influence, likely from upwelling dust, was found in the extratropical transition layer in spring. Eruptions of both extratropical and tropical volcanoes that injected aerosol into the stratosphere impacted the stratospheric aerosol load for up to a year if

their clouds reached lower than 20 km altitudes, whereas deeper reaching tropical injections rose in the tropical pipe and impacted it for several years. Over the years 2006-2015, volcanic eruptions increased the stratospheric AOD on average by ~40%. In absolute numbers the stratospheric AOD and radiative forcing amounted to 0.008 and -0.2 Wm$^{-2}$, respectively.

## 1 Introduction

Aerosol particles have a large impact on the Earth's climate. Trends in the abundance of aerosol particles remain as an

important component in the climate system of which influence on climate is still highly uncertain (IPCC, 2014). For example, volcanic eruptions inject particles and the sulfate forming precursor gas sulfur dioxide ($SO_2$) into the stratosphere, where particles may remain for up to several years. There, the aerosol particles scatter and absorb solar radiation hindering it from heating the surface of the Earth. The eruption of Mount Pinatubo in 1991 resulted in a global cooling in excess of 0.5°C in the following years (McCormick et al., 1995)

In recent years much attention (Fyfe et al., 2016; Rajaratnam et al., 2015; Trenberth, 2015; Xie et al., 2016; Yan et al., 2016) has been drawn to a discrepancy between the Earth system models' (CMIP5) projections of temperatures to that of an observed slow-down in the warming trend in the beginning of this century (Fyfe et al., 2013), often called the hiatus. Part of the discrepancy was caused by bias in the temperature data (Karl et al., 2015). Likely reasons behind the remaining

discrepancy are fluctuations in oceanic heat sequestration, solar blocking by volcanic aerosol in the stratosphere and solar



forcing not captured by the state of the art models (Andersson et al., 2015; Medhaug et al., 2017; Meehl and Teng, 2014; Myhre et al., 2013; Santer et al., 2014; Solomon et al., 2011).

In the past decade several volcanic eruptions perturbed the stratospheric aerosol load, causing periods of increased

stratospheric aerosol optical depth (AOD), that has been observed via remote sensing from satellites and ground (Khaykin et al., 2017; Sakai et al., 2016; Vernier et al., 2011; Zuev et al., 2017), and in-situ observations (Martinsson et al., 2017).

In the tropics, upwelling tropospheric air enters the stratosphere, bringing aerosol particles and precursor gases including $SO_2$ and carbonyl sulfide (OCS), that form the background stratospheric aerosol. It is still debated which of these

components that dominates as source (Brühl et al., 2012; Sheng et al., 2015). While the water soluble $SO_2$ may be scavenged by cloud processes in convective transport, a large fraction of the OCS gets transported to the stratosphere where it oxidizes by UV-radiation at ~25 km altitude forming the Junge layer (Crutzen, 1976). A most recent study by Rollins et al. (2017) indicates that $SO_2$ concentrations in the tropical UT are too low for general tropical $SO_2$ upwelling to be a significant source of the stratospheric background aerosol.

Sulfate is the dominating constituent in the background as well as in the volcanic aerosol (Deshler, 2008). In addition, organics (Friberg et al., 2014; Murphy et al., 2014), meteoritic matter, soil dust, and volcanic ash can be found in stratospheric aerosol particles. The ash component decreases readily after a volcanic eruption as the large ash particles get gravitationally separated from the sulfate forming volcanic $SO_2$ layers and subside out of the stratosphere. It thus generally

constitutes only a small fraction of the stratospheric aerosol.

Sandwiched between the tropopause and the overlying 380 K isentrope, the LMS constitutes the lowest lying portion of the stratosphere. It is connected with the troposphere through isentropic surfaces crossing the tropopause, where significant bidirectional transport of air takes place (Holton et al., 1995). Local air-exchange occurs in a region extending a few

kilometers above the tropopause, termed the extra-tropical transition layer (ExTL; Hoor et al., 2002). It is characterized by steep gradients in chemical species that have different concentration levels in the troposphere and stratosphere (Gettelman et al., 2011).

The Brewer-Dobson circulation transports air meridionally from the tropical stratosphere, in an overturning circulation

(Gettelman et al., 1997). Most of the air is transported in a low altitude, or shallow, branch that spans an isentropic range of 380 K to ~450-470 K (Fueglistaler et al., 2009; Lin and Fu, 2013). A portion of the air ascends in the so-called tropical pipe. This high-altitude, or deep, branch extends to the stratopause (~50 km altitude) and transports the air slowly, resulting in residence times of several years (Bönisch et al., 2009). Air is then subsiding through the midlatitude and polar stratosphere,



bringing aerosol particles down to the LMS from higher stratospheric altitudes that eventually end up in the troposphere, where it is readily scavenged by cloud processes.

The CALIOP instrument onboard the CALIPSO satellite is based on the use of a lidar to retrieve observations of the
backscattering from aerosol particles and clouds at high vertical resolution (Winker et al., 2007). Vernier et al. (2009) developed a method of filtering out clouds, to study the backscattering from aerosol particles above 15 km altitude. The stratospheric AOD have been computed from the lidar backscattering from particles by assumptions on particle size distribution and composition (Jäger and Deshler, 2002, 2003). Solomon et al. (2011) estimated the radiative forcing induced by the injection of volcanic aerosol above 15 km altitude during year 2000 - 2010 to be on average ~-0.1 Wm$^{-2}$. Their
approach excluded the lower-lying LMS, which contains approximately 40% of the stratospheric mass. Including the LMS, Ridley et al. (2014) and Andersson et al. (2015) revealed that the lowermost stratosphere (LMS) contains a significant fraction of the stratospheric aerosol, which influence on the AOD had been neglected in previous studies. Hence, for estimation of the full climate impact of volcanism one needs to consider the whole stratospheric column.

In this paper we present the volcanic influence on the "entire" stratospheric AOD (from the tropopause to 35 km altitude) over time and space, starting with studies on the transport of volcanic aerosol within the stratosphere. New techniques of handling the CALIOP data will be presented and discussed: one being the removal of signals from polar stratospheric clouds (PSC) and the other a means of correcting data in periods when the lidar signal gets attenuated by dense aerosol layers. Finally, the regional and global AODs are presented for the entire stratosphere in relation to transport patterns. As a means to
represent the residence time of the aerosol from eruptions at several latitudes, reaching various altitudes, the AOD is presented for three stratospheric layers: the LMS, the potential temperature range of 380 to 470 K, and for altitudes above the 470 K isentrope.

## 2 The CALIOP data

This study of the stratospheric aerosol in the time period 2006 – 2015 is based on measurements with the CALIOP (Cloud-
Aerosol Lidar with Orthogonal Polarization) instrument aboard the satellite CALIPSO (Cloud-Aerosol Lidar and Infrared Pathfinder Satellite Observation) in a joint mission of NASA and the French space agency, CNES. The almost nadir-viewing (3°) CALIOP is a two-wavelength lidar utilizing three receiver channels (Winker et al., 2007, 2009). One channel measures backscatter intensity at 1064 nm and two channels measure orthogonally polarized backscatter intensity at 532 nm. The measurements result in high resolution vertical profiles of aerosols and clouds: in the altitude ranges 8.2 – 20.2, 20.2 – 30.1
and 30.1 – 40 km the vertical resolutions are 60, 180 and 300 m, respectively. During one day CALIPSO performs approximately 15 orbits between 82°S and 82°N with a repeat cycle of 16 days (Winker et al., 2010).



This work is based on the Level 1 (version 4-10) night-time products of the 532 nm perpendicular and parallel polarized channels. The total from these two channels provides the backscatter intensity. Their ratio, the polarization ratio is used to infer the shape of the scattering objects. In the present study each swath measured by CALIOP was averaged over 1° in
latitude and averaged or interpolated to 180 m in altitude. The polarization ratio was used to identify pixels containing clouds. From that a mask was formed for removal of pixels containing signals from clouds, using a 5% threshold in the polarization ratio (Vernier et al., 2009). The mask was expanded to remove weak signals from the edges of clouds, and data below thick clouds were excluded (Andersson et al., 2015); compare Figure 1b with Figure 1a for an example.

The measured backscattering intensity is the sum of that by aerosol particles and air molecules. In order to separate these two components, the molecular scattering of the air was modeled based on ozone number density, atmospheric temperature and pressure from the Global Modeling and Assimilation Office (GMAO; http://gmao.gsfc.nasa.gov/). In this study we use the ratio of the total scattering ($\beta$) to the modeled molecular scattering ($\beta_m$), the so-called scattering ratio (SR), and the difference between them, the aerosol scattering (AS):

$$SR = \frac{\beta}{\beta_m} \tag{1}$$

$$AS = \beta - \beta_m = (SR - 1) \cdot \beta_m \tag{2}$$

The SR is an optical equivalent to mixing ratio and the AS to aerosol concentration.

The measured backscattering intensity is really an 'attenuated backscattering' ($\beta'$), due to light extinction of the laser pulses caused by molecules and particles. The attenuation that the laser pulses experience while passing through the atmosphere is estimated using the so-called two-way transmission parameter ($T^2$), which is the product of the two-way transmissions by the two components (molecules and particles). The molecular part is estimated from modelling data of the molecular
background (Winker et al., 2009; Young et al., 2005), while the influence of the other generally is considered negligible (e.g. Khaykin et al., 2017; Vernier et al., 2009). We will discuss this in section 5, along with a method we developed to correct for the increased attenuation occurring during periods when volcanism increased the stratospheric aerosol load.

**3. Computing the stratospheric AOD**

In order to obtain the AOD from the CALIOP lidar measurements at 532 nm wavelength the aerosol particle size distribution
is needed. It is well established that large volcanic eruptions like the Mt Pinatubo in 1991 cause a shift towards larger particle sizes in the stratosphere (Deshler, 2008), resulting in a variable relation between lidar backscatter and extinction



(Jäger and Deshler, 2003). Measurements in the LMS show that volcanic eruptions of Grimsvötn and Nabro in 2011 only had minor influence on the particle size distribution (Martinsson et al., 2014), and they were similar to size distributions obtained in periods of small volcanic impact (Jäger and Deshler, 2002). Therefore, the background size distribution reported by the latter authors was used in this work. The AOD as a function of latitude was obtained by integrating the AS in the

vertical direction, and multiplying with the so-called lidar ratio, i.e. the extinction to backscattering ratio. We use a lidar ratio of 50 sr provided by Jäger and Deshler (2003). That value corresponds to the background stratospheric aerosol concentration and size distribution occurring in the end of the 1990s. In a recent study Prata et al. (2017) find lidar ratios after volcanic eruptions, of Kasatochi (2008), Sarychev (2009) and Puyehue-Cordón Caulle (2011), to be somewhat higher but not significantly differing from the factor 50 sr, reported by Jäger and Deshler (2003).

Existing stratospheric AOD datasets excludes the LMS. They are based on measurements where the AOD was integrated with the lower altitude limit being either the 380 K isentrope (Bourassa et al., 2012; Sato et al., 1993) or 15 km altitude (Solomon et al., 2011; Vernier et al., 2011). A first AOD estimation based on high resolution measurements that included the LMS were presented by Andersson et al. (2015). As the lower altitude limit of the AOD integration they used the tropopause

supplied with the CALIOP data set, a static tropopause (Pan and Munchak, 2011) based on GEOS5 (Goddard Earth Observing System Model Version 5) data. The thermal tropopause according to the WMO definition on average resides more than 1 km above the dynamic tropopause in the extra-tropics (Wilcox et al., 2012).

In the present study we use a dynamic tropopause as it captures the chemistry better than the thermal one does. This is
reflected in that part of the volcanic aerosol in the present study resided below the thermal tropopause. The location of the dynamic tropopause was computed from potential vorticity (PV) values obtained from ERA-Interim reanalysis data provided by the ECMWF (European Center for Medium Ranged Weather Forecasts), and averaged to the same resolution as was the CALIOP data (1° latitudinally, 180 m vertically). Using a lower tropopause will of course increase the stratospheric AOD and the duration of effects on stratospheric AOD from volcanic eruptions. It will thereby provide a better estimate of the

total stratospheric AOD. However, lowering the tropopause leads to more influence from the ExTL (Hoor et al., 2004), thus increasing the influence from tropospheric aerosol on the estimate of the stratospheric AOD. The location of the dynamic tropopause will be discussed more in section 6.1.2.

## 4 Handling PSCs

The presence of PSCs sometimes results in strong back-scattering signals in the winter polar stratosphere that can cause bias
to the stratospheric AOD. This is especially problematic in the Southern Hemisphere where the PSCs occur at latitudes from 60°S. This is illustrated in Figure 1a as the high SR extending up to 25 km altitude in the Antarctic region. For investigation





of the volcanic influence on the stratospheric aerosol, the influence from PSCs needs to be negligible, requiring a means of excluding data affected by these clouds. Andersson et al. (2015) solved this problem by manually omitting periods with PSCs. In the present study, we present a general approach to exclude the PSCs from the aerosol data.

Since PSC formation requires temperatures below 195 K, that temperature was used as a minimum threshold for a PSC mask applied to the Polar Regions, as means of minimizing the bias from PSCs. Furthermore, data below the clouds were removed to avoid bias from the strongly attenuated signals below the optically dense PSCs. This automated approach resulted in an almost complete removal of PSC signals, as illustrated by comparing Fig 1b and c, while keeping almost all of the data. Weak PSC signals remain, but its influence become negligible when averaged globally or hemispherically since the
Antarctic region (60-90°S) constitutes only 6% of the Earth's surface area.

## 5 Correcting for errors caused by particle extinction

### 5.1 Attenuation of the laser

Passing through the atmosphere, the laser gets attenuated by scattering and absorption from aerosol particles, molecules and $O_3$. Hence, the CALIOP instrument retrieves an 'attenuated backscattering' from laser pulses. Computing the true
backscattering requires information on the extinction through the overlying atmosphere. The attenuation is accounted for by the two-way transmission ($T^2$), where an expression for the corrected backscattering becomes

$$\beta = \frac{\beta'}{T^2} \tag{3}$$

where $\beta'$ is the attenuated backscattering retrieved by CALIOP. The two-way transmission depends on two attenuating components, i.e. that of molecules ($T_m^2$) and particles ($T_p^2$) (Young et al., 2005), and can be expressed as their product:

$$T^2 = T_m^2 \cdot T_p^2 \tag{4}$$

where the absorption from ozone molecules here is included in $T_m^2$. By combining equations 1, 3 and 4, an expression for the SR corrected from both molecular extinction and particles becomes:

$$SR_{mpC} = \frac{\beta'}{T_m^2 \cdot T_p^2 \cdot \beta_m} \tag{5}$$



Attenuation from molecules is computed based on modelling, but that by particles are generally not considered in studies based on CALIOP data. For example, Vernier et al. (2009) and Khaykin et al. (2017) discuss that the attenuation from particles is less than 1% at 15 km altitude in absence of strong volcanic eruptions, arguing that corrections are unnecessary during these time periods. Assuming $T_p^2 = 1$, the expression for the SR corrected from molecular extinction becomes:

$$SR_{mC} = \frac{\beta'}{T_m^2 \cdot \beta_m}$$ (6)

from which the molecular extinction corrected AS ($AS_{mC}$) can be computed using eq.2.

10     Volcanic clouds may however result in non-negligible attenuation of the lidar signals. By neglecting the attenuation caused by particles, part of the aerosol signal will be accounted as signal coming from molecules, because the latter signal is also attenuated by the aerosol. Unaccounted for, the attenuation by volcanic aerosol particles can result in underestimation of the full effect of volcanic impact of the stratospheric aerosol load and the corresponding AOD. The attenuation is naturally increasing as the laser beams (and scattered light) passes through the atmosphere causing the largest errors to occur for 15     signals retrieved from the lower altitude side of volcanic clouds, and below them. In the present work this corresponds to the strongest attenuation being at the tropopause and in the UT.

In the present study, we first corrected the data by computing the two-way transmission caused by the attenuation from molecules (eq.6). This correction is sufficient in the absence of volcanic clouds, but light extinction caused by aerosol 20     particles in dense volcanic clouds resulted in further attenuation of the CALIOP laser. This is evident in Fig 2 (black dashed line) where apparent decreases in SR arise in the UT after the volcanic eruptions of Sarychev, Nabro and Calbuco. The most dramatic decrease is observed for the SR data after the eruption of Calbuco in 2015, where strong attenuation resulted in unphysically low SR, i.e. below 1.

25    **5.2. Estimating the two-way transmission from particle extinction**

The two-way transmission from particles can be retrieved by a complicated technique where strongly attenuating features in the CALIPSO data are first identified and then an iterative process is used as means of estimating particle extinctions in attenuating layers (Hostetler et al., 2006). The layer detection requires the use of a threshold value for identification of attenuating features. The resulting particle extinction parameter in the CALIOP Level 2 data is compared with the AS in Fig. 30     3. The algorithm obviously fails to detect most of the relatively fresh volcanic clouds (compare Fig 3a with 3b), and when the volcanic aerosol is mixed with the background aerosol 5 months after the eruption (Fig 3c and d) the Level 2 procedure



does not detect any attenuating aerosol. In Fig. 2b it is clear that the SR in the UT still is strangely lowered 5 months after the Calbuco eruption. We therefore developed a means of correcting for the attenuation caused by volcanic aerosol particles, where the particle related two-way transmission ($T_p^2$) is calculated and applied on the $SR_{mC}$ data. This procedure follows below.

The apparent decrease in the UT SR (Fig 2) following the largest eruptions of the time period studied indicates attenuation of the laser signal induced by particles. By assuming the UT conditions to remain approximately unchanged after volcanic eruptions the degree of attenuation in a given volcanically perturbed time and place can be estimated by comparison with the signals in the years without strong volcanic influence. However, a volcanically perturbed stratosphere results in a small

10   elevation of the UT aerosol concentration. That effect can be assumed small for most of the eruptions of this study, but the lower cloud of Kasatochi had a direct impact on the UT aerosol load (Andersson et al., 2015). In section 5.2.1 we present a method to handle that eruption separately. By assuming the UT conditions to remain approximately unchanged after all but the Kasatochi eruption, we can estimate the two-way transmission from particles at the altitude of the tropopause

$$\langle SR_{ut}\rangle_v = \langle SR_{ut}\rangle_{bg} \Rightarrow \frac{\langle SR_{mC,ut}\rangle_v}{\langle T_{p,ut}^2\rangle_v} = \frac{\langle SR_{mC,ut}\rangle_{bg}}{\langle T_{p,ut}^2\rangle_{bg}} \Rightarrow \frac{\langle T_{p,tp}^2\rangle_v}{\langle T_{p,tp}^2\rangle_{bg}} = \frac{\langle SR_{mC,ut}\rangle_v}{\langle SR_{mC,ut}\rangle_{bg}}$$

where $T_{p,tp}^2$ is the two-way transmission at the tropopause, and the indexes $v$ and $bg$ represent volcanic and background conditions. Furthermore, we assume that $\langle T_{p,tp}^2\rangle_{bg}=1$ since the particle extinction is small in background conditions, and we end up with

20   $$\langle T_{p,tp}^2\rangle_v = \frac{\langle SR_{mC,ut}\rangle_v}{\langle SR_{mC,ut}\rangle_{bg}} \tag{7}$$

It is obvious that $T^2$ values theoretically can be in the range of 0 to 1. In this work $\langle T_{p,tp}^2\rangle_v$ are distributed from 0.95 to 1, with the majority of values being >0.97. Thus, the decreases in SR under volcanic clouds are mostly on the orders of 2-5%, which may appear to be negligibly low. For example, Fig 2a shows typical decreases in SR to be on the order of 0.025 in absolute

25   terms, from for example $\langle SR_{mC,ut}\rangle_{bg}$ of 1.050 to $\langle SR_{mC,ut}\rangle_v$ of 1.025. Using eq.7 these values correspond to a $\langle T_{p,tp}^2\rangle_v$ of 0.976. However, transferred to the AS and AOD these deviations become relevant. Using eq. 2, and assuming $\beta m$ equal during volcanic periods and background, we end up with

$$\frac{\langle AS_{mC,ut}\rangle_v}{\langle AS_{mC,ut}\rangle_{bg}} = \frac{\langle SR_{mC,ut}\rangle_v - 1}{\langle SR_{mC,ut}\rangle_{bg} - 1} = \frac{1.025 - 1}{1.050 - 1} = 0.50$$





the $\langle AS_{mC,ut} \rangle_v$ and $\langle AS_{mC,ut} \rangle_{bg}$ being the respective AS in the UT during volcanic and background conditions. Thus, if the attenuation by particles is unaccounted for, the AOD in the UT gets underestimated by as much as 50% in the current example, since we retrieve the AOD linearly from the AS. A small effect on the SR transforms into a large effect on AS and AOD because before the correction half of the aerosol signal was accounted as signal from molecules in the example above.

This simple example illustrates that apparent tiny decreases in the $T^2$ may result in large underestimations in the computed AOD of low altitude layers, i.e. the LMS and especially the ExTL and UT. Once computed, the $\langle T^2_{p,tp} \rangle_v$ are combined with the column integrated $AS_{mC}$ data to compute two-way transmission matrices for the whole stratospheric column ($T^2_p$).

To compute $T^2_p$ values for the whole stratospheric column, the integrated column of stratospheric $AS_{mC}$ values are used as a
means of normalizing the $T^2_p$ values from the 'top of the atmosphere', here 35 km altitude, down to the tropopause. In this procedure the uppermost altitude bin gets a $T^2_p$ value of 1, altitude bins below get lower $T^2_p$ values, down to the tropopause where the $T^2_{p,tp}$ is computed from eq.7. We start by using the fraction that gets extinct by particles

$$Ext_p = 1 - T_p \tag{8}$$

and formulate an expression for estimating the fraction extinct in a single altitude bin (*a*)

$$Ext_{p,a} = \frac{AS_{mC,a}}{\sum_{TP}^{TOA} AS_{mC}(z)} \cdot Ext_{p,tp} \tag{9}$$

where $Ext_{p,tp}$ is the extinct fraction at the tropopause $(1 - T_{p,tp})$, $AS_{mC,a}$ is the molecular corrected AS in altitude bin *a*, and the denominator holds the integrated backscattering for the total stratospheric column where *z* is the altitude bin. The transmission through the column can be computed by multiplying the transmission through each layer. Such approach may result in instabilities when the $AS_{mC}$ values become close to zero or negative, as were the case after the eruption of Calbuco, i.e. the $SR_{mC} < 1$ in Fig 2b corresponds to negative $AS_{mC}$. By instead assuming linearity we can compute the extinction at
each altitude simply by summing the extinction in each overlying layer. Combined with eq.8, this procedure yields the following expression for the altitude dependent two-way transmission from particles:

$$T^2_p(z) = \left\{ 1 - \frac{\sum_{z+1}^{TOA} AS_{mC}(z)}{\sum_{TP}^{TOA} AS_{mC}(z)} \left( 1 - T_{p,tp} \right) \right\}^2 \tag{10}$$

where *z* is the altitude bin, and *TOA* is the 'top of the atmosphere' (35 km altitude). This linear method produces $T^2_p$ values that in the current study deviate by << 1% from those computed by the multiplicative method, and the resulting AOD values



deviate by <1%. However, the deviations grow with the size of the particle extinction values and may become more relevant for eruptions of the size of the 1991 Pinatubo eruption.

### 5.2.1 The Kasatochi eruption

Since Kasatochi's lower cloud increased the UT aerosol load the $\langle T_{p,tp}^2 \rangle_v$ could not be retrieved directly using eq.7 for that eruption. Instead we made use of the Sarychev eruption. For Sarychev we found the extinct fraction $(1 - \langle T_{p,tp} \rangle_{Sar})$ to be related to the amount of aerosol in the column, i.e. the column integrated AS ($\sum_{TP}^{TOA} AS_{Sar}$). The $\langle T_{p,tp}^2 \rangle_v$ after Kasatochi was estimated indirectly from that relation:

$$\frac{\sum_{TP}^{TOA} AS_{Kas}}{1-\langle T_{p,tp} \rangle_{Kas}} = \frac{\sum_{TP}^{TOA} AS_{Sar}}{1-\langle T_{p,tp} \rangle_{Sar}} \tag{11}$$

where the indexes indicate the respective volcano. Combined with eq.8 and rearranging yields the expression for estimating the UT particle related two-way transmission after the Kasatochi eruption

$$\langle T_{p,tp}^2 \rangle_{Kas} = \left\{ 1 - \left( \frac{\sum_{TP}^{TOA} AS_{Kas}}{\sum_{TP}^{TOA} AS_{Sar}} \right) \cdot \langle T_{p,tp} \rangle_{Sar} \right\}^2 \tag{12}$$

### 5.3. Correcting the data

Once the $T_p^2$ is known, the scattering ratios can be corrected in regard to the attenuation by particles by combining equations 5 and 6

$$SR_{mpC} = \frac{\beta'}{T_m^2 \cdot T_p^2 \cdot \beta_m} = \frac{SR_{mC}}{T_p^2} \tag{13}$$

and the corresponding aerosol scattering ($AS_{mpC}$) can be computed from insertion into eq.2. By this, our method introduces a simple means of correcting the data to account for particle attenuation.

To study the sensitivity of the method, an iterative process was performed where the $AS_{mpC}$ data were used to compute new $T_p^2$ values. We found only negligible changes in the $SR_{mpC}$ and $AS_{mpC}$ after these iterations. Thus, the developed procedure was found to be a robust method for correcting stratospheric CALIOP data for the attenuation caused by particle extinction.



Correcting the CALIOP data following volcanic eruptions, according to the equations above, significantly changes the SR as shown in Figure 2 (dashed vs. full lines). The global stratospheric AOD in the 8 first months after eruptions increases by the percentages 4 (Kasatochi), 6 (Sarychev), 6 (Nabro), and Calbuco 7% (Fig 2b). These numbers are the result of relatively small volcanic elevations in the stratospheric aerosol load. The influence of particle extinction naturally grows with the strength in elevation of the aerosol concentrations. Thus, eruptions of the size of the 1991 eruption of Mt Pinatubo, which was more than a magnitude larger than the eruptions of this study, would result in a strong need of correcting the data for the particle extinction.

## 6 Results

Large quantities of background sulfuric acid aerosol are produced deep into the stratosphere from OCS that is transported from the tropical troposphere in the Brewer-Dobson circulation (Crutzen, 1976) together with particles, $SO_2$ and other particle precursors from natural and anthropogenic sources. To this rather stable stratospheric background, special events inject large amounts of particles and precursor gases into the stratosphere, causing large variability in the stratospheric aerosol load. The most common cause of this variability is explosive volcanic eruptions (Robock, 2000), but also large fires can occasionally influence the stratospheric aerosol (Fromm et al., 2010). For the time period studied here, mid-2006 to 2015, several volcanic eruptions and two large fires had such a potential. Table 1 lists these events starting more than a year before mid-2006 in order to account for the long residence time of volcanic aerosol going deep into the stratosphere, e.g. the aerosol load following the Mt Pinatubo eruption in 1991 declined over several years (McCormick et al., 1995).

### 6.1 The volcanic impact on the SR and AS

The stratospheric aerosol load varied substantially during the studied period as indicated in Figure 4, where the SR is illustrated in relation to latitude and altitude averaged for the months January, April, July and October over the years 2006 - 2015. Already in the first month, July 2006, a feature is seen in the tropics at 19 km altitude. This was caused by the eruption of the Soufriere Hills volcano in May 20, 2006 (Vernier et al., 2009), approximately one month before the start of measurements from CALIPSO. In October the same year a second feature appears to the south and at slightly lower altitude, as a result of the eruption of Rabaul in October 7, 2006. The next major feature appears in October 2008. Aerosol from the eruption of the extratropical volcano Kasatochi in August 7, 2008 formed two layers at different altitudes. The influence from this eruption remained in January 2009, and in the frame for April more volcanic aerosol was added close to the tropopause by the eruptions of Redoubt in March and April 2009. A feature appearing in the southern tropics in April has been identified as smoke from bush fires in February 2009 in Victoria, Australia (Vernier et al., 2011). In July 2009 the Northern Hemispheric stratosphere was perturbed again, this time by the extratropical volcano Sarychev (June 12, 2009), with the SR signal clearly visible until Jan 2010. In January 2011 aerosol from the eruption of the tropical volcano Merapi



(November 5, 2010) had affected mainly the Southern Hemispheric stratosphere. The SR of three volcanic eruptions can be observed in July 2011. The Southern Hemispheric Puyehue-Cordón Caulle (June 5, 2011) and the Northern Hemispheric Grimsvötn (May 21, 2011) affected the respective LMS, and aerosol from the tropical Nabro (June 12, 2011) went to the north and affected the stratosphere during approximately one year. After a couple of years with lower activity the tropical

volcano Kelut erupted (February 13, 2014), reaching above 20 km altitude. The majority of the SR signal was confined to the tropics, but over a year, part of the aerosol was transported to the extratropics while the remaining tropical aerosol rose to higher altitudes. Finally, the Southern Hemispheric extratropical volcano Calbuco (April 23, 2015), induced the highest SR that occurred in the Southern Hemisphere during 2006-2015, which effect still remain at the end of the period.

Another interesting feature in Fig 4 is the variability at altitudes above 30 km, which is connected with the Quasi-Biennial Oscillation (QBO) (Vernier et al., 2011). The stratospheric temperature varies with the QBO, in the 30 – 50 hPa layer by almost 10°C, with the highest temperatures in the westerly shear (Baldwin et al., 2001). Modeling indicates that the QBO-associated temperature variability induces QBO-related altitude dependence in sulfuric acid evaporation from the aerosol (Hommel et al., 2015). Hence, at these altitudes less amount of aerosol is present during the westerly shear.

The SR deals with properties relative to the air mass, but from climatic point of view the AS, i.e. the absolute measure of the scattering by aerosols (eq.2), is of importance. Figure 5 shows the AS as a function of time and altitude in six latitude bands, each constituting 16% of the earth's surface area. The white lines mark the mean upper and lower limits of the LMS, i.e. the 380 K isentrope and the tropopause.

Elevated AS are clearly observed both after the stronger and weaker volcanic eruptions, and a striking feature in figure 5 is the transport patterns where volcanic clouds in the tropic (extra-tropics) ascend (descend). Several eruptions reached above the 380 K, but it is also shown that a large portion of the aerosol was located in the LMS. Part of that aerosol came from the direct injection of volcanic clouds in the extratropics, and due to the latitudinal transport within the Brewer-Dobson

circulation and subsiding in the midlatitudes, also the aerosol from tropical eruptions eventually ended up in the LMS. Furthermore, a seasonal variation is revealed in the strong signal from particles in the extra-tropical troposphere with the maximum AS occurring in spring/summer. A small part of this tropospheric source extends into the ExTL. Its possible influence on the stratospheric aerosol load will be discussed in section 6.3.

## 6.2 Using the dynamic tropopause

The dynamic tropopause was chosen to represent the lower boundary of the LMS as it is expected to best enclose the stratospheric air carrying volcanic aerosol. The PV of the dynamic tropopause varies between different studies (Gettelman et al., 2011), and is generally considered to lie somewhere in the range of 1.5 – 3.5 PVU (Hoerling et al., 1991; Hoinka, 1997;

Kunz et al., 2011). The PV fields become vertical close to the equator. Therefore the 380 K isentrope is used as a limit of the dynamic tropopause's maximum possible altitude. Thus, the altitude of the dynamic tropopause becomes the lowest of the 380 K isentrope and that of a chosen PV-surface. The lowest commonly used dynamic tropopause, the 1.5 PVU surface, is on average located approximately 1.3 km below the thermal tropopause. Integration from this dynamic tropopause increases

the total stratospheric AOD compared to when using the thermal tropopause. Here the aim is to use as low PV level as possible for best possible enclosure of the stratospheric volcanic aerosol. However, before deciding the tropopause level we need to consider the impact from tropospheric aerosol in the ExTL.

In Figs 1 to 5, we assumed the 1.5 PVU level to be a good representation of the tropopause. Figure 6 illustrates a simple

means of investigating the best suited PV value of a dynamic tropopause where the SR in approximately 0.25 km thick layers of the atmosphere above the 1.5 PVU level (LMS) are plotted for the midlatitudes of each hemisphere over the entire time frame of our study.

It is evident from the Fig 6 that the strong volcanic eruptions of Sarychev and Calbuco induced strong gradients in the SR

throughout the ExTL, and that volcanic influence is present down to the lowest PV-range, i.e. the 1.5-2 PVU. This is further corroborated by in situ particulate sulfur measurements (Martinsson et al., 2017). Unlike these eruptions, Kasatochi injected large amounts of volcanic aerosol in the lower parts of the LMS and the UT. As a result, the aerosol signal is strong also in the lowest LMS layer of Fig. 6a.

The local tropospheric sources' contribution to the stratospheric aerosol load are small in comparison to the signals of strong volcanic eruptions such as those from Kasatochi, Sarychev or Calbuco, making them negligible in the perspective of the stratosphere's total AOD. They do however have significant relative influence on the ExTL in the absence of strong volcanic eruptions.

Based on the investigation above, a PV-value of 1.5 PVU was chosen to represent the dynamic tropopause in the following analyses, as it best captures the "full" volcanic impact on the stratosphere and on the climate.

### 6.3 Volcanic and tropospheric impact in the ExTL

Many interesting features can be seen in comparisons of the atmospheric slices in the LMS (fig 6) and of the two hemispheres. The largest peaks in the SR occurred after major volcanic eruptions in the extratropics (Kasatochi, Sarychev

and Calbuco), which affected its respective hemisphere. In the aftermath, the SR is highest deep into the LMS (at the highest PVs) and decreases down to the tropopause. This gradient is caused by gradual mixing of stratospheric air, carrying volcanic aerosol, with cleaner tropospheric air.



In the absence of major volcanic eruptions, the highest SR is found in spring close to the tropopause. Thus, the SR gradient is reversed compared to the periods dominated by volcanism, and it is more evident in the Northern than in the Southern Hemisphere. Figure 5 shows that the gradient is connected to a spring-/summertime increase in AS in the troposphere.

Hence, we conclude that tropospheric local sources have significant influence in the ExTL during spring. These sources are obviously less important in the Southern Hemisphere. Air-craft measurements in the ExTL revealed that upwelling dust peaks in spring (Martinsson et al., 2005). These observations indicate that the spring/summer peaks in the lowest part of the LMS are caused by upwelling dust.

## 7 Discussions

The stratospheric AOD is obtained by converting aerosol scattering to extinction based on the particle size distribution and chemical composition (Jäger and Deshler, 2002, 2003), and integrating in the vertical direction. The patterns of volcanism are evident in Figure 7 where we divided the stratosphere into three layers for which we calculated AOD.. This sub-division was based on transport patterns: the upper layer extends down to the 470 K isentrope, and represents the region were the latitudinal transport is weak (Fueglistaler et al., 2009; Lin and Fu, 2013), the LMS constitute the lowest layer, and in

between is the mid-layer spanning isentropes of 380 – 470 K, where the shallow Brewer-Dobson branch is strong. This categorization of stratospheric layers will be used in the following discussions.

The temporal trends of the global and hemispheric mean AODs are compared in Figure 8 for the three layers and for that of the "entire" stratosphere. Even though the LMS mostly is confined to the extratropics, and that altitudes above 470 K

constitute a small portion of the stratospheric mass, it is evident that aerosol in these layers make up a significant portion of the global stratospheric AOD. In times of low volcanic impact the three layers contribute in approximately equal amounts to the total stratospheric AOD. Furthermore, Fig 7 and 8 show that the global stratospheric AOD reached its lowest values in the studied decade around the year 2013. This is in agreement with in-situ observations that find the stratospheric aerosol load to be at background conditions in 2013 (Martinsson et al., 2017).

The influence from spring-time dust in the ExTL (section 6.3; Fig 6) on the total stratospheric aerosol AOD is very small (Fig 8). Thus, fluctuations caused by local tropospheric sources (upwelling dust) have only minor influence on the AOD estimation, and, in that respect, validates once again that the 1.5 PVU level is a well suited tropopause for studying the influence from volcanism.

We further explore the AOD distribution by investigating the strong latitudinal patterns shown in Figure 7. This is illustrated in Fig 9, where the AOD was averaged over different time periods, starting with the entire decade in Fig 9a. There we find the AOD to be distributed very differently in the respective layers. The upper layer has high AOD in the tropics that decreases to the poles. The LMS naturally shows highest AODs in the extratropics. The largest elevations are observed in the

Northern Hemisphere due to the volcanic influence being stronger there. The mid-layer is shown to have more evenly distributed AOD, with slightly higher AOD in the extra-tropics, and highest in the Northern Hemisphere. In figure 9b – d we separate the layers, to compare the influence from three types of volcanic eruptions to that of the decadal mean and that of the background conditions (year 2013).

## 7.1 The different eruption types

It is evident that the impact on the stratospheric AOD varied between eruptions and that only few eruptions reached into the uppermost layer. The eruptions that had significant impact on the stratospheric AOD were grouped into three categories depending on the observations in Fig 7. These types are as follows:

Trop I - Tropical eruptions with deep-reaching volcanic clouds, which were to a large degree incorporated into the deep
Brewer-Dobson branch and ascended in the tropical pipe.

Trop II - Tropical eruptions with clouds that were confined in the mid-layer, transported in the shallow Brewer-Dobson branch, and not significantly transported in the deep branch.

Extrop - Volcanic clouds from extratropical eruptions. Some of these were confined within the LMS, and others partly penetrated into the mid-layer where the aerosol was incorporated into the shallow Brewer-Dobson branch and spread to the tropics.

The impact of the volcanic eruptions will be discussed based on these categories in the following sections along with the two
wildfires.

### 7.1.1 Deep reaching tropical volcanic eruptions

Two volcanoes significantly affected the upper layer, i.e Soufrierre Hills (May 2006) and Kelut (Feb 2014). In addition, part of the elevations in AOD from summer 2006 may have come from the Jan 2005 Manam eruption (Vernier et al., 2009). The high reaching part of the Soufrierre Hills cloud slowly ascended in the tropics. A strong elevation of the AOD is visible for
more than one and a half years after the eruption after which the AOD decreases slowly over the following years until reaching its lowest values in 2013, before the Kelut eruption. Kelut exploded after a period of stratospheric background




levels, making it easy to track the spread of the volcanic aerosol. A large fraction of the volcanic cloud slowly ascended with the deep Brewer-Dobson branch in the tropical pipe, whereas the lower part of it was spread latitudinally in the mid-layer, mostly to the southern extratropics (Figs 4 and 9). The high-altitude part of the cloud was confined within the tropics. A small sudden increase in the AOD is observed in the southern extratropics approximately one and a half years after the Kelut

eruption. That elevation was likely connected with the midlatitude eruption of Calbuco (Apr 2015). Minor fluctuations in the upper layer's aerosol load may have been caused by evaporation/condensation of sulfuric acid (Vernier et al., 2011) in the upper part of the layer in connection with temperature variability induced by the quasi-biennial oscillation (Hommel et al., 2015). The aerosol in the upper layer is eventually transported out to the next lower layer at midlatitudes, i.e the one located between the 380-470 K isentropes.

## 7.1.2 Tropical eruptions below the 470 K isentrope

The volcanic clouds of the tropical eruptions from Rabaul (Oct 2006), Merapi (Nov 2010) and Nabro (Jun 2011) all penetrated the tropopause but did not reach altitudes above the 470 K isentrope (~20 km). While the two former eruptions impacted both hemispheres, the Nabro eruption mostly influenced the Northern Hemisphere (Fig 7 and 9). The small increase shown after Nabro in the Southern Hemisphere LMS (Fig 9) was caused by the extratropical eruption of Puyehue-

Cordón Caulle (Jun 2011). The aerosol from the Trop II eruptions was incorporated into the shallow Brewer-Dobson branch and spread to the midlatitudes within weeks. There the volcanic aerosol subsided and increased the AOD of the LMS, while decreasing in the mid-layer (Fig 7 and 8). The AOD in the mid-layer decreased over ~9 months in the case of the strong Nabro eruption, and the subsidence through the LMS resulted in several months prolonging of the volcanic impact of the eruption (Fig 7 and 8).

## 7.1.3 Extra-tropical volcanic clouds

Several extra-tropical eruptions impacted the Northern Hemispheric stratosphere, and some influenced the southern one. In the Northern Hemisphere, Kasatochi (Aug 2008) and Sarychev (Jun 2009) induced the strongest elevations in the AOD. Part of the aerosol from these eruptions reached the mid-layer, where it was spread to the tropical stratosphere within the shallow Brewer-Dobson branch. Fig 8 shows that some patterns of the global and Northern Hemispheric mean AOD in the mid-layer

after Sarychev are similar to that of the Nabro eruption. The latitudinal distributions in the mid-layer shows similarities (Fig 9c), but being a tropical eruption Nabro naturally impacted more in the tropics.

The Kasatochi eruption formed two clouds. Kasatochi's dense lower cloud was confined to the LMS and UT. Its stratospheric part was transported to the tropopause within approximately three months. The remaining weaker signal comes

from the upper cloud that descended to the LMS (Fig 8). The upper cloud reached the 380 – 470 K layer, and shows some similarities with that from the Sarychev and Nabro eruptions, having a rapid latidudinal transport in the lower BD branch.



For all three eruptions, the elevation of the AOD remained until the following spring until the strong subsidence transported the aerosol down to the troposphere.

The Calbuco eruption (Apr 2015) was by far the largest one in the Southern Hemisphere. The volcanic cloud reached above the LMS, and induced a rapid strong elevation of the AOD of the mid-layer. The subsiding aerosol increased the AOD of the LMS so that it peaked a few months later than in the mid-layer. Calbuco reached higher altitudes than Kasatochi and Sarychev did, explaining the slower transport down to the LMS.

There were also a number of minor influences from extratropical volcanic eruptions that are shown as small increases in Figures 7 and 8, e.g. the eruptions of Redoubt (Mar 2009), Grimsvötn (May 2011) and Puyehue-Cordón Caulle (Jun 2011). Their clouds contained lower amounts of aerosol and did not penetrate as deep into the stratosphere, as the extratropical eruptions discussed above. Hence, the aerosol was rapidly transported out of the stratosphere, similarly to the lower cloud following the Kasatochi eruption.

### 7.1.4 The forest fires

The two forest fires affecting the stratosphere in this 10 year period had significantly lower impact on the stratospheric AOD than the volcanic eruptions. However, the fire in February 2009 reached altitudes of more than 20 km (fig 4 and 7a). A concurrent small increase in the AOD is shown in the southern tropics upper layer (fig 7a), that mixed with the volcanic aerosol from the Soufrierre Hills eruption. This fire therefore could have made a small, but long-term impact on the stratospheric AOD. The fire in December 2006 affected mainly the southern LMS, as evidenced in Fig. 6c, with duration of a few months.

### 7.2 Patterns of volcanic aerosol in the stratosphere

Summarizing the findings of volcanic perturbations and transport within the stratosphere we make the following observations:

1.  Aerosol from extratropical eruptions injected into the LMS remains there until transported out to the troposphere, e.g. the lower Kasatochi (2008) cloud. The AOD gets elevated during less than a year.

2.  Volcanic injections close to the extratropical tropopause only briefly impacts the stratospheric aerosol load, for example Grimsvötn (May 2011).

3.  Aerosol from extratropical eruptions reaching the mid-layer gets dispersed hemispherically, and will be found at higher altitude in the tropics than their extratropical injection altitude, but no clear case of interhemispheric





exchange. Examples are the upper Kasatochi cloud, the upper part of the Sarychev (2009) cloud and Calbuco (2015).

4.   Volcanic clouds from tropical eruptions reaching above the 470 K isentrope (approximately 20 km altitude) tend to move upwards without a strong poleward transport, e.g. Soufriere Hills (2006) and Kelut (2014). Hence, the aerosol remains in the stratosphere for several years.

5.   Volcanic clouds injected close to the tropical tropopause tend primarily to be transported poleward and within 1 – 2 months reach midlatitudes, e.g. Rabaul (2006), Merapi (2010) and Nabro (2011). The Nabro eruption shows that the AOD gets increased during up to approximately one year. For the other two eruptions the signals are too weak for such estimations.

6.   The last, clear indications of influence from extratropical as well as tropical volcanic eruptions appear in the LMS at mid- and high latitudes, as manifested by the three largest eruptions in this study (according to Table 1), i.e. Kasatochi, Sarychev and Nabro.

These observations agree well with the large-scale circulation pattern in the stratosphere. The general circulation of the stratosphere (Brewer-Dobson) is directed upwards in the tropics and downwards in the extratropics. As a consequence volcanic clouds injected into the LMS, which is not isentropically connected with other parts of the stratosphere, are transported downwards to the troposphere (point 1), and clouds injected close to the extratropical tropopause are rapidly removed due to that transport (2). Volcanic clouds reaching above approximately 20 km altitude in the tropics (4) get incorporated in the upper Brewer-Dobson branch and become relatively isolated from the extratropics for years (Fueglistaler et al., 2009), whereas tropical eruptions reaching lower altitudes are rapidly mixed meridionally (5) via the shallow Brewer-Dobson branch. This also explains the observation of reverse transport of extratropical volcanic clouds where aerosol reaching above the upper boundary of the LMS (3) is transported in the shallow Brewer-Dobson branch. Finally, the observation that the volcanic clouds of both tropical and extratropical eruptions leave the stratosphere via the LMS (6) is connected with the direction of the Brewer-Dobson circulation. Obviously, volcanic aerosol that reached the stratosphere ends up in the LMS sooner or later, no matter the injection latitude or altitude of the stratospheric injection.

### 7.3 Climate relevance

Comparison of the mean AOD for 2006 – 2015 to that in the background (the year 2013) reveals that the volcanic eruptions increased the stratospheric AOD by ~40% (Table 2). The eruptions of the extratropical Sarychev and tropical Nabro volcanoes both increased the AOD with ~70% over the course of a year. The higher-reaching Kelut eruption had a smaller initial influence, i.e. ~20% in the first year after eruption. It is expected to impact the stratospheric AOD over several years due to the slow transport in the deep Brewer-Dobson branch. Hence, averaged over longer time-spans the Kelut eruption likely causes an impact on the stratospheric AOD of similar size as the more sulfur-rich eruptions of Sarychev and Nabro.



The resulting radiative forcing was estimated as in Hansen et al. (2005) and Solomon et al. (2011), using a conversion factor of -25 for AOD to radiative forcing, is added to figure 8 as a secondary y-axis. The global radiative forcing from stratospheric aerosol ranged from approximately -0.15 in the background to -0.35 Wm$^{-2}$ after the strongest volcanic eruptions

of the period (Fig 8). Most of the elevation in radiative forcing appeared in the Northern Hemisphere, and the lowest occurring during the year 2013. On average the global stratospheric radiative forcing amounted to -0.2 Wm$^{-2}$ during a time-period of a decade (Table 2).

**8 Conclusions**

We present a study on the stratospheric aerosol optical depth (AOD) and radiative forcing over a period of almost a decade

(mid 2006 - 2015), covering periods of varying volcanic impact as well as stratospheric background conditions, with a resolution of 1° latitudinally and 8 days timewise. This required development of new methods in order to prevent influence from polar stratospheric clouds, and to correct data when the lidar was attenuated by volcanic aerosol. The latter correction increased the AOD by 4-7% in the first year after the volcanic eruptions of Kasatochi (2008), Sarychev (2009), Nabro (2011) and Calbuco (2015).

Strong volcanic impact was found in the extra-tropical tropopause layer (ExTL) down to potential vorticities (PV) of 1.5 PVU, i.e. more than 1 km below the thermal tropopause. We therefore used the 1.5 PVU level as the tropopause in our analysis to include the full impact of volcanism. In spring/summer upwelling dust clearly elevated the aerosol load, up to PV-levels of 5-6 PVU, i.e. ~2 km above the dynamic tropopause, but it had insignificant influence on the total AOD of the

lowermost stratosphere (LMS).

The stratospheric AOD was studied by dividing the stratosphere into three layers that incidentally carry approximately the same global AOD during conditions close to the stratospheric background; the LMS, the altitude range between the 380 and 470 K isentropes, and altitudes above the 470 K isentrope (~20 – 35 km). Several eruptions were found to influence the two

lower layers, both extratropical and tropical ones.

Only the high-reaching aerosol from the tropical volcanoes Soufriere Hills (May 2006) and Kelut (Feb 2014) clearly impacted the upper layer (>470 K). Their volcanic clouds were first observed at altitudes of ~20 km, after which the clouds rose to higher altitudes incorporated in the deep Brewer-Dobson branch, impacting on the stratospheric AOD over several

years. After the Soufriere Hills eruption the aerosol gradually decreased over the following years, which also has been



observed for the far stronger Pinatubo eruption (1991). A similar decay was observed after the Kelut eruption, up till the end of the period studied here.

Volcanic clouds reaching into the mid-layer (380 - 470K) were not found to rise, but spread latitudinally in the shallow Brewer-Dobson branch, before being transported down to the LMS and eventually out of the stratosphere in the extratropics. For example, aerosol from the eruptions of the extratropical volcanoes Sarychev (Jun 2009) and Calbuco (Apr 2015) spread to the tropics within weeks, whereas aerosol from the eruption of the tropical volcano Nabro (Jun 2011) spread in the opposite direction followed by subsiding to the LMS. Such transport was limited within a Hemisphere and impacted the stratosphere for up to a year.

The stratospheric AOD was elevated the most in the extratropics, due to the combined effect of latitudinal transport and the larger stratospheric column. The majority of that elevation came from aerosol located in the LMS. Subsidence through the LMS causes the AOD to remain elevated in the LMS for several months after that the overlying stratosphere has returned to its background aerosol levels.

We have included the LMS in an estimation of the AOD of the "entire" stratosphere over the period 2006 – 2015. The stratospheric background AOD and the volcanic impact were found to be ~50% higher in midlatitudes than in the tropics. Volcanism was found to have elevated the average global stratospheric AOD by ~40%. The stratospheric aerosol had a cooling effect of the Earth, which in terms of radiative forcing, is estimated to -0.2 $Wm^{-2}$.

**Acknowledgements**

Financial support from the Swedish National Space Board (contract 130/15) and the Swedish Research Council for Environment, Agricultural Sciences and Spatial Planning (contract 942-2015-995) is gratefully acknowledged. Aerosol products from the CALIOP sensor were produced by NASA Langley Research Center.

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

**Tables**

10    **Table 1. Volcanic eruptions and wild fires in the 21st century that affect (or had the potential to affect) the aerosol loading of the stratosphere**

| Volcano | | Date | Lat. | Long. | VEI[†] | SO$_2$ (Tg) |
|---------|---|------|------|-------|-----|----------|
| Ulawun | Ul | 29 Sep. 2000 | 5° S | 151° E | 4 | * |
| Sheveluch | Sh | 22 May 2001 | 57° N | 161° E | 4 | * |
| Ruang | Ru | 25 Sep. 2002 | 2° N | 125° E | 4 | 0.03[a] |
| Reventador | Ra | 3 Nov. 2002 | 0° S | 78° W | 4 | 0.07[a] |
| Anatahan | At | 10 May 2003 | 16° N | 146° E | 3 | 0.03[a] |
| Manam | Ma | 27 Jan. 2005 | 4° S | 145° E | 4 | 0.09[a] |
| Sierra Negra | Si | 22 Oct. 2005 | 1° S | 91° W | 3 | * |
| Soufrière Hills | So | 20 May 2006 | 17° N | 62° W | 3 | 0.2[b] |
| Rabaul | Rb | 7 Oct. 2006 | 4° S | 152° E | 4 | 0.2[a] |
| Jebel at Tair | Je | 30 Sep. 2007 | 16° N | 42° E | 3 | 0.08[c] |
| Great Divides Fire | Gd | 1 Dec. 2006 | 37° S | 144° E | - | - |
| Chaitén | Ch | 2 May 2008 | 43° S | 73° W | 4 | 0.01[d] |
| Okmok | Ok | 12 Jul. 2008 | 53° N | 168° W | 4 | 0.1[c] |
| Kasatochi | Ka | 7 Aug. 2008 | 52° N | 176° W | 4 | 1.7[c] |
| Fire in Victoria | Vi | 7 Feb. 2009 | 37° S | 145° E | - | - |
| Redoubt | Re | 23 Mar. 2009 | 60° N | 153° W | 3 | 0.01[e] |
| Sarychev | Sa | 12 Jun. 2009 | 48° N | 153° E | 4 | 1.2[f] |
| Eyjafjallajökull | Ey | 14 Apr. 2010 | 64° N | 20° W | 4 | * |
| Merapi | Me | 5 Nov. 2010 | 8° S | 110° E | 4 | 0.4[g] |
| Grimsvötn | Gr | 21 May 2011 | 64° N | 17° W | 4 | 0.4[h] |
| Puyehue-Cordón Caulle | Pu | 6 Jun. 2011 | 41° S | 72° W | 5 | 0.3[h] |
| Nabro | Na | 12 Jun. 2011 | 13° N | 42° E | 4 | 1.5[h] |
| Kelut | Ke | 13 Feb. 2014 | 8° S | 112° E | 4 | 0.2[i] |
| Calbuco | Ca | 23 Apr. 2015 | 41° S | 73° W | 4 | 0.3[j] |



† Volcanic Explosivity Index (from Global Volcanism Program (http://www.volcano.si.edu/))

* Not available

[a] Prata and Bernardo (2007)

[b] Carn and Prata (2010)

5  [c] Thomas et al. (2011)

[d] Carn et al. (2009)

[e] Lopez et al. (2013)

[f] Haywood et al. (2010)

[g] Surono et al. (2012)

10  [h] Clarisse et al. (2012)

[i] Li et al. (2017)

[j] Pardini  et al.(2017)

**Table 2. Mean global stratospheric AOD and radiative forcing for years 2006-2015, the year 2013 and the first year after the**
15  **volcanic eruptions in Fig 9.**

| Period | AOD | RF (Wm$^{-2}$) |
|---|---|---|
| Years 2006 - 2015 | 0.0082 | -0.21 |
| The year 2013 | 0.0059 | -0.15 |
| Sarychev +1year | 0.0102 | -0.26 |
| Nabro +1year | 0.0099 | -0.25 |
| Kelut +1year | 0.0073 | -0.18 |

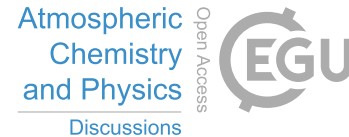

**Figures**

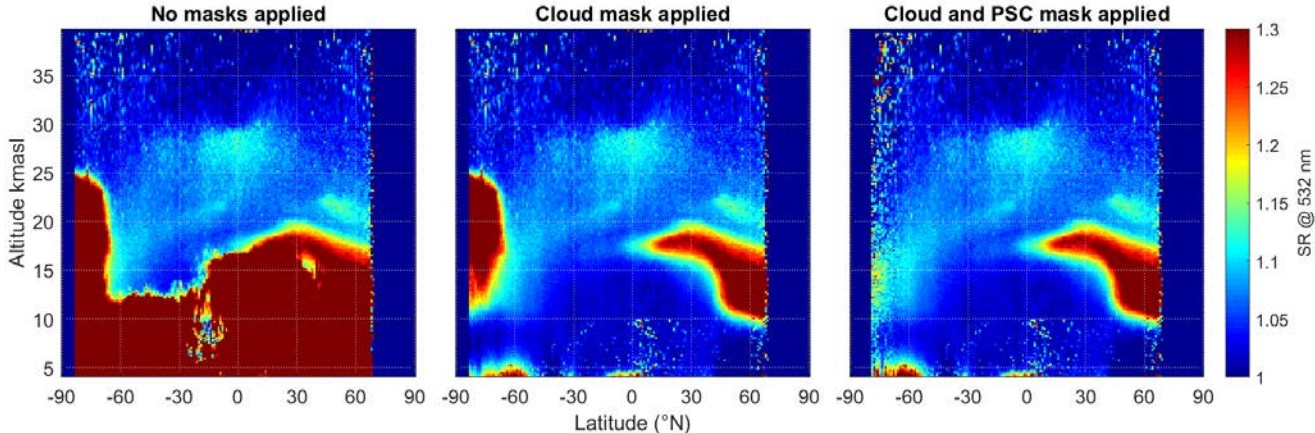

**Figure 1. Illustration of the removal of cloud signals from the CALIOP data. (a) The original data, (b) and (c) removal of signals from ice clouds and polar stratospheric clouds, respectively. Data averaged over Aug 2009 were used in this example. The strong aerosol signal in the Northern Hemisphere comes from aerosol connected to the Sarychev eruption. The white lines mark the average altitudes of the tropopause and the 380 K isentrope.**



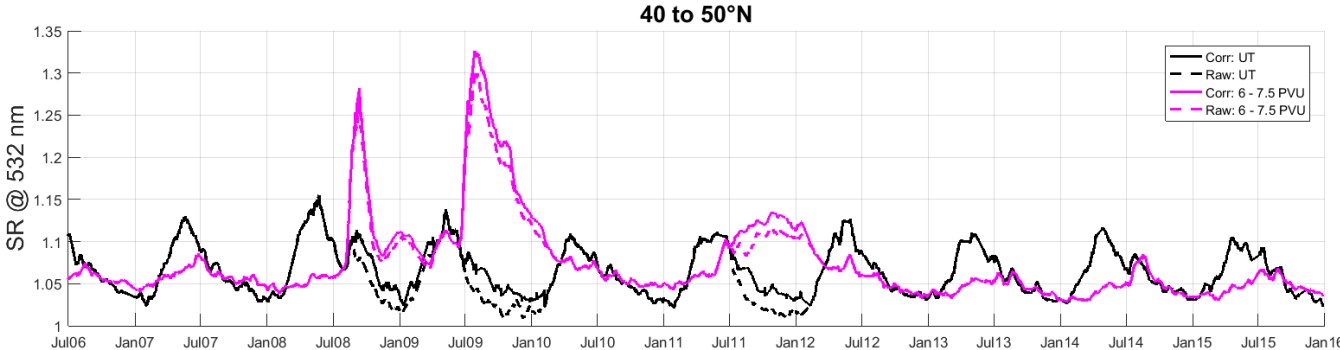

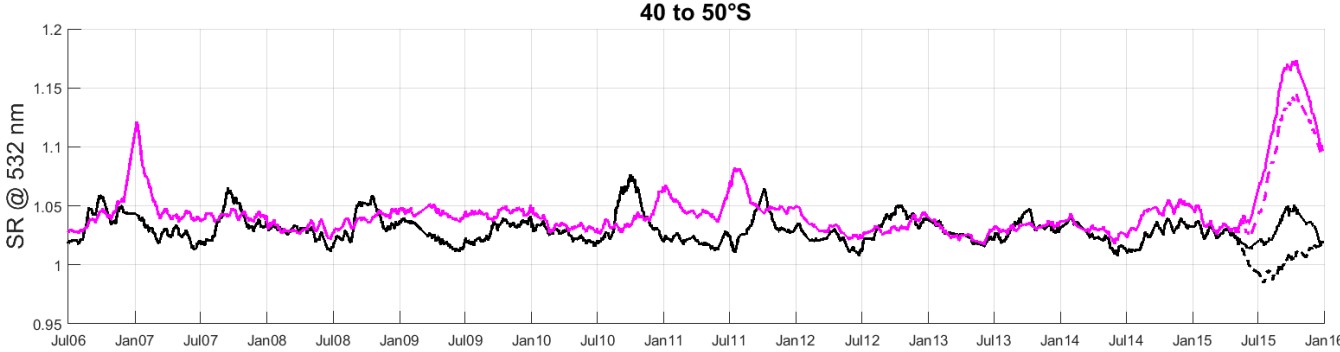

**Figure 2. Examples illustrating the effects from corrections made to compensate for the laser attenuation caused by aerosol particles. The scattering ratios (SR) were averaged over the latitude intervals a) 40-50°N and b) 40-50°S for the UT (black; 250 - 1000 m below the tropopause) and the PV-layer of 6-7.5 PVU (magenta). Both layers are on average approximately 750 m thick). Corrected (Uncorrected) data are marked as full (dashed) lines.**





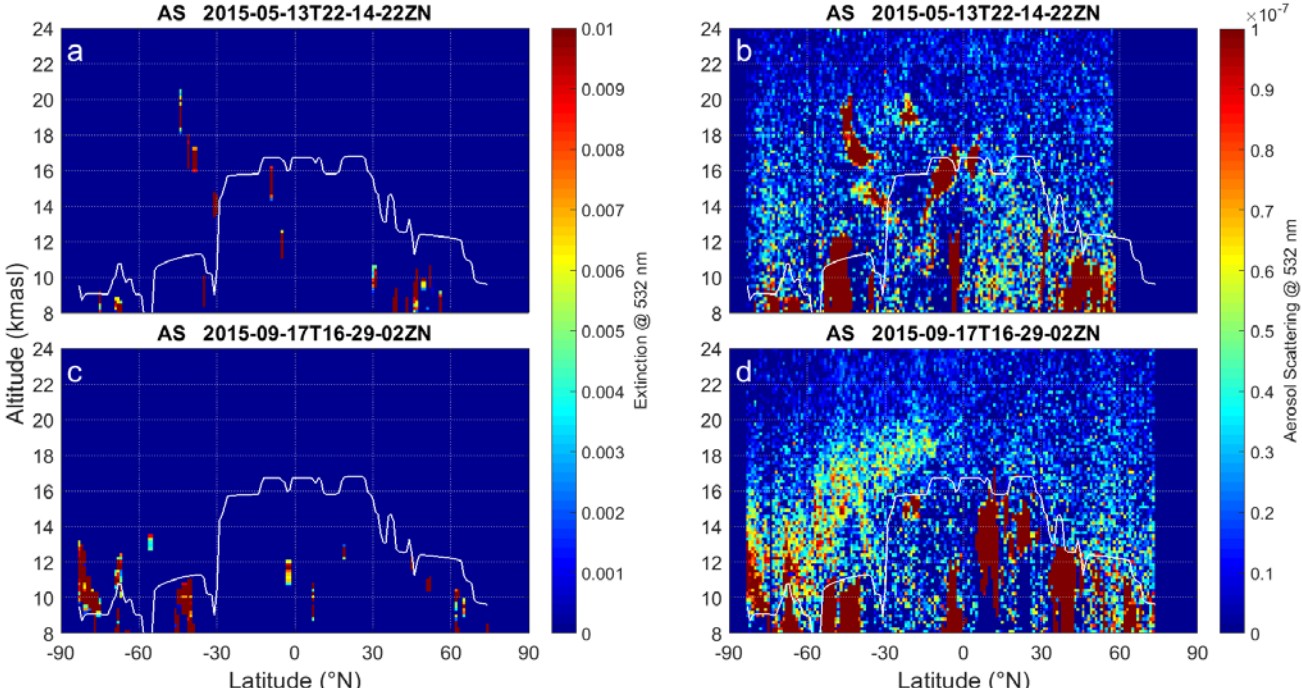

**Figure 3. Comparisons of the particle extinction parameter from the level 2 data v4-10 (a and c), and the aerosol scattering (b and d) taken after the Calbuco eruption. The upper figures were taken three weeks after (on May 13, 2015), and the lower almost four months after (on Sep 17, 2015) the eruption. Data in b and d were based on the attenuated backscattering at 532 nm in the level 1 v4-10 data, and modeling of the molecular scattering. The white lines mark the static tropopause.**





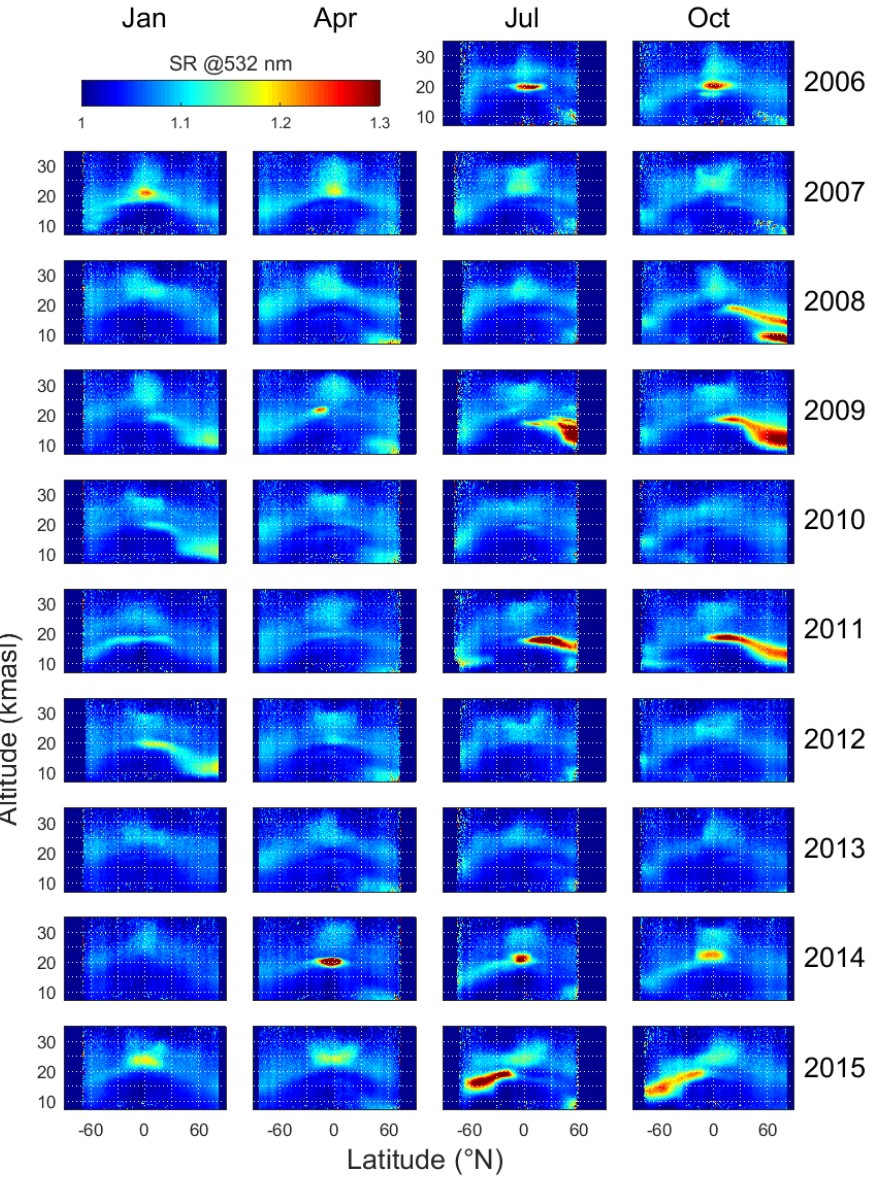

**Figure 4. Monthly mean scattering ratio as a function of latitude and altitude (7 - 35 km) for years 2006 – 2015 and months January, April, July and October.**

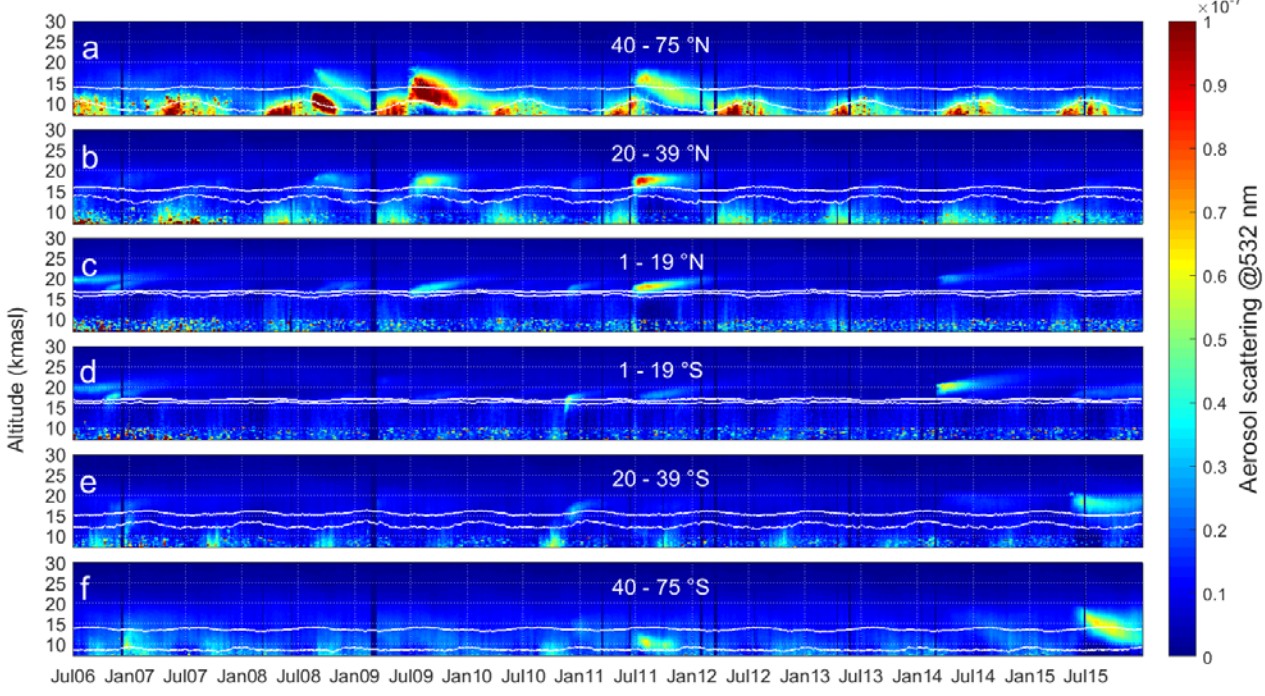

**Figure 5. Aerosol scattering against time and altitude in six latitude bands each covering 16% of the earth's surface area.**

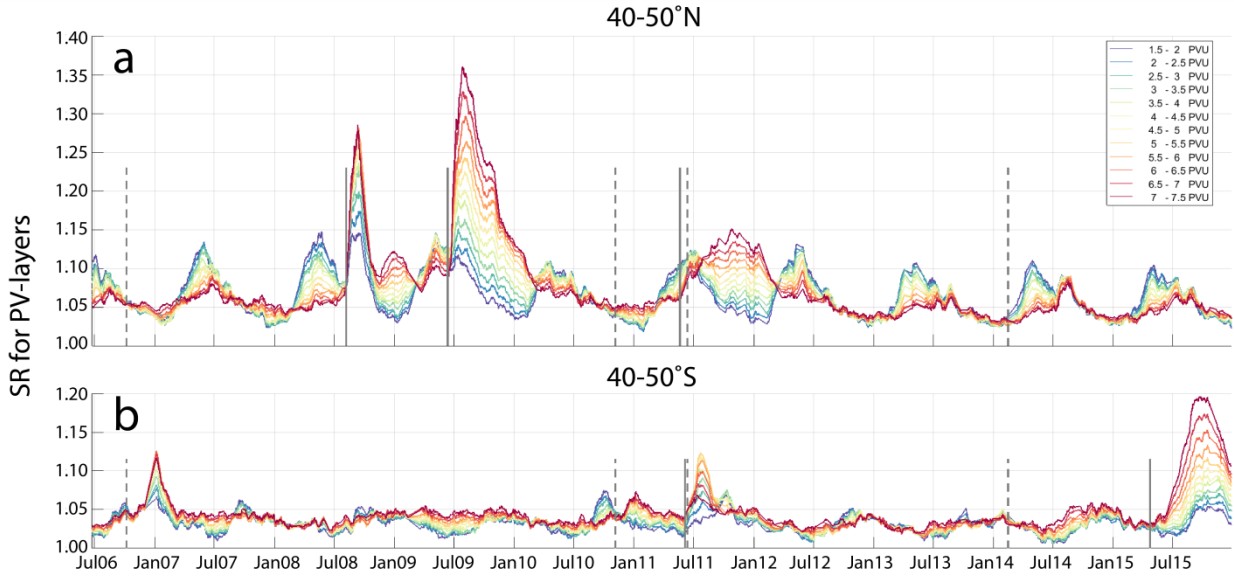

5  **Figure 6. Temporal trends of the scattering ratio in altitude layers in the LMS (PV intervals of 0.5 PVU), for the latitude bands 40-50°N (a) and 40-50°S (b). Vertical dashed lines marks tropical volcanic eruptions, and dashed indicates extratropical eruptions that impacted the respective Hemisphere.**



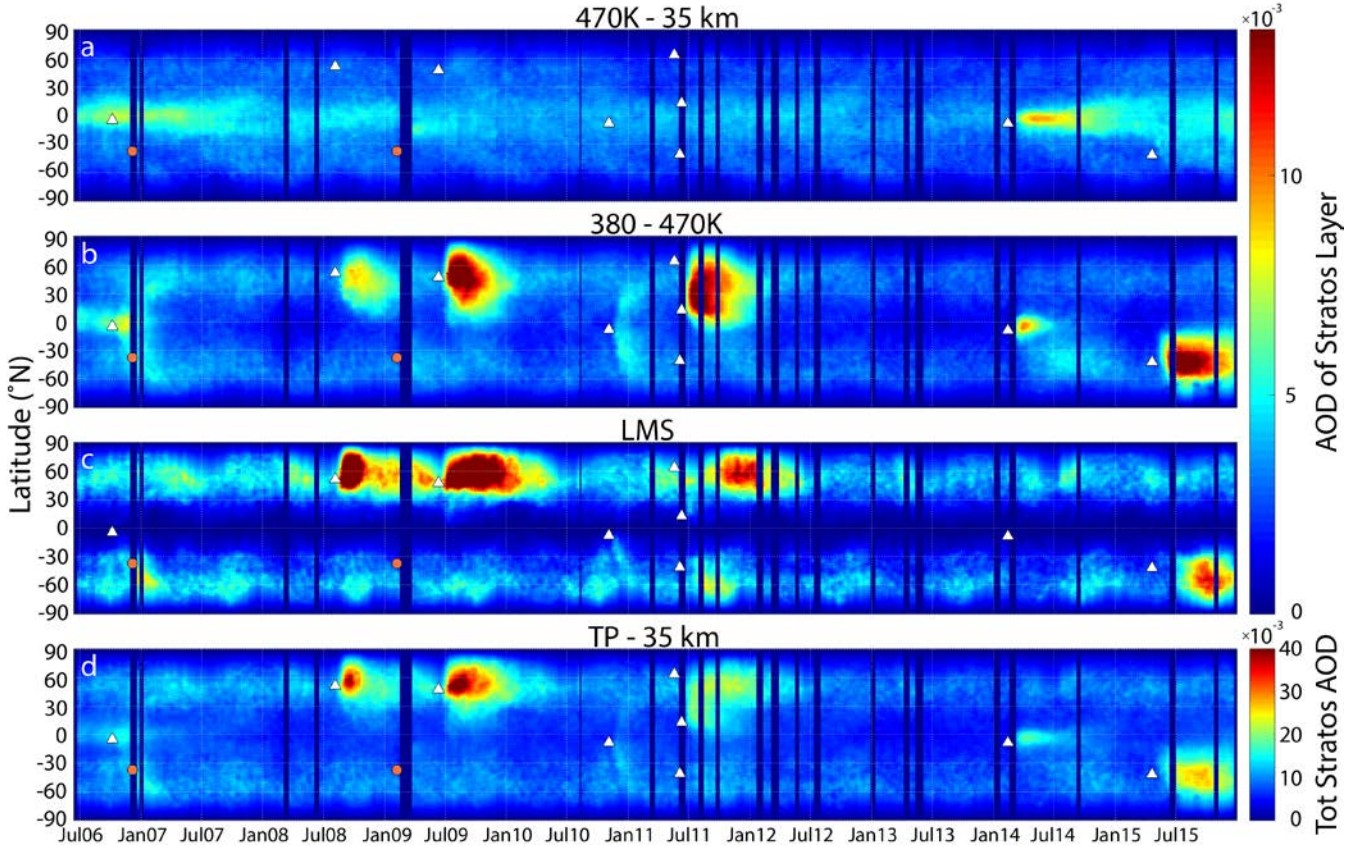

**Figure 7. The latitudinal distribution over ten years for the latitude weighted stratospheric AOD: a) above the 470 K isentrope, b) between the 380 and 470 K isentropes, c) the LMS, and d) that of the entire stratosphere. The color bar for a-c spans a range of 1/3 of that in d. Triangles (circles) marks time and latitude of relevant volcanic eruptions (forest fires).**




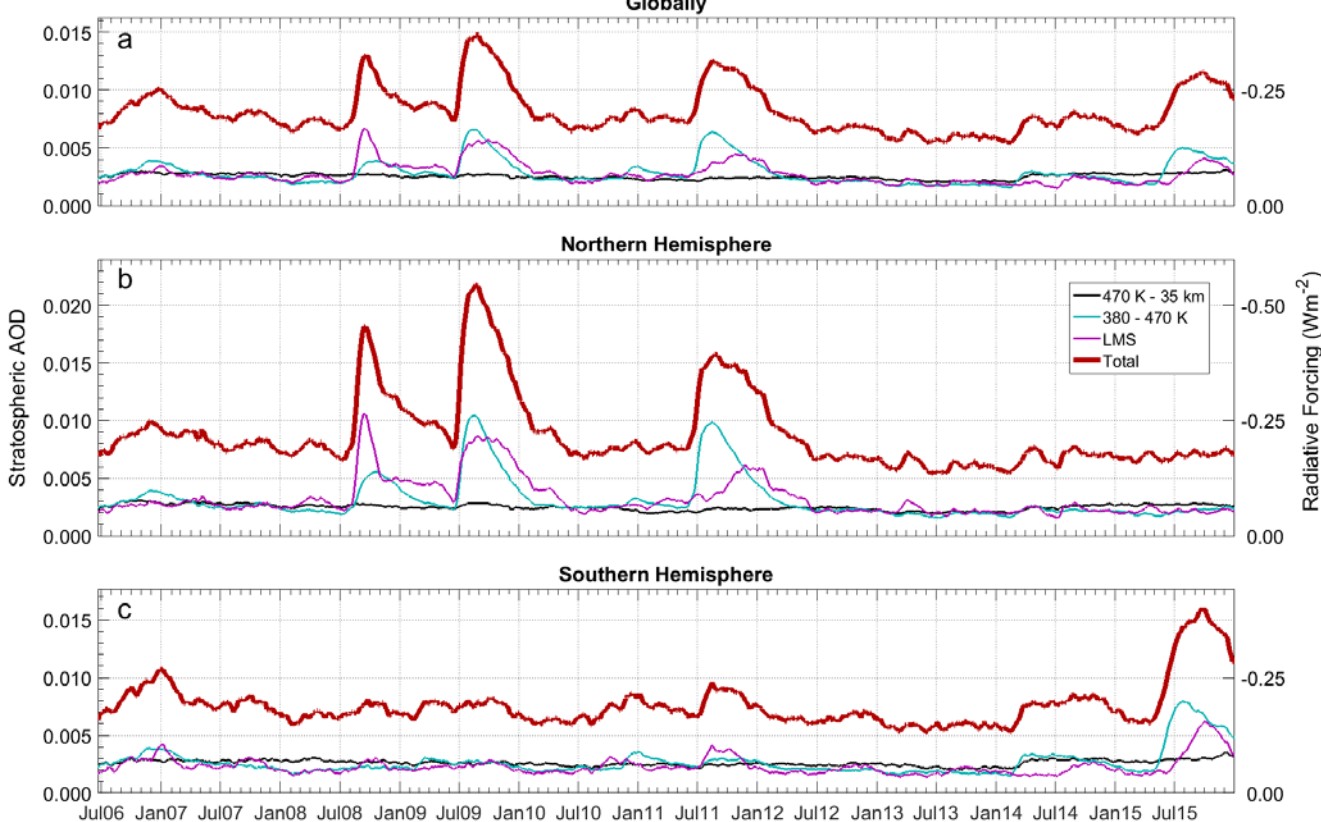

**Figure 8. Temporal trends of the stratospheric AOD and radiative forcing for a) the entire globe, b) the Northern Hemisphere, and c) the Southern Hemisphere. The lines mark the AOD of the LMS (magenta), altitudes between the 380 and 470 K isentropes (cyan), altitudes above the 470 K isentrope (black), and the entire stratosphere (thick red). Note the reversed scale for the radiative forcing on the right y-axis. Data were extrapolated to the Polar Regions and latitude weighted before averaging.**





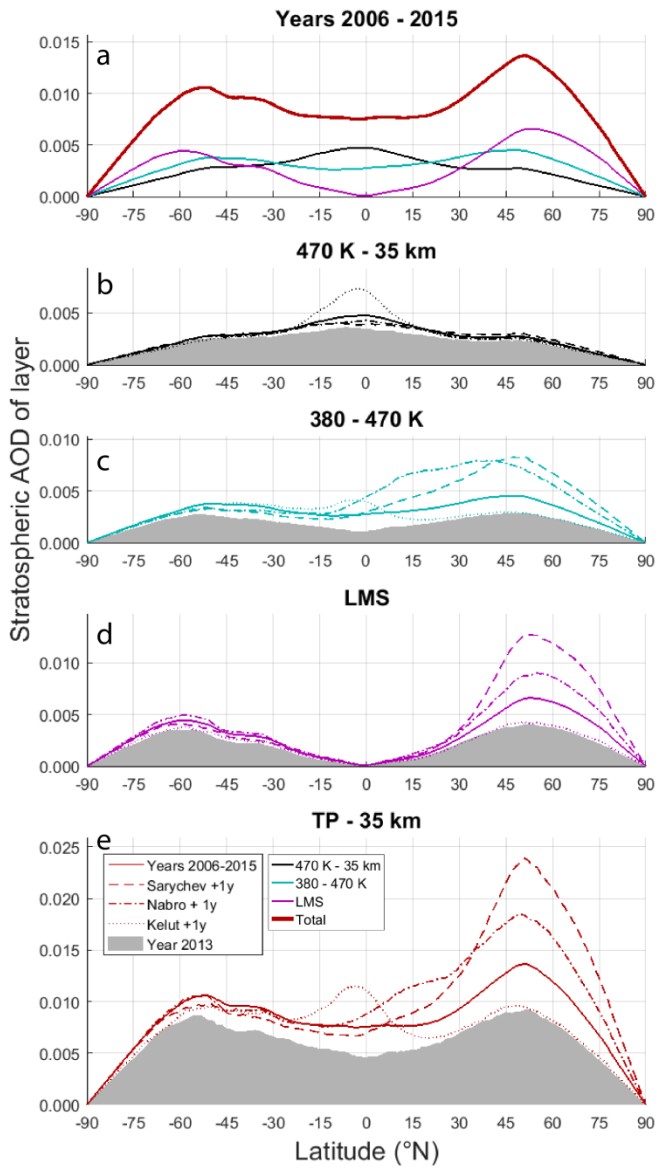

**Figure 9. The latitudinal distribution of the latitude weighted AOD. a) The AOD of the different stratospheric layers and of the total stratospheric AOD (thick red) for the entire period (2006-2015). b-d) shows the respective layers and e) the AOD of the entire stratospheric column, for the time periods noted in the legend. Lines represent AODs in the altitude intervals 470 K to 35 km (black), 380 – 470 K (cyan), the LMS (magenta), and the entire stratosphere (thick red). 2013 (grey filling) is considered as background conditions, and the line styles mark the different periods as noted in the legend.**