# Peer review of "Volcanic impact on the climate – the stratospheric aerosol load in the period 2006 – 2015"

_Atmospheric Chemistry and Physics, 2017_

## Referee Comment (RC1) · Anonymous Referee #1 · 17 Mar 2018

This contribution to our understanding of stratospheric aerosol optical depth (AOD) focuses on the CALIOP backscatter data set after it is cleared of clouds. A significant fraction of the paper is devoted to a procedure to correct CALIOP measurements beneath volcanic clouds when the laser is attenuated by the volcanic cloud. The procedure is relatively straight forward, although I found the discussion of how the threshold is set to instigate the procedure, and the relevance of Figure 3 to that procedure, confusing. I don't have specific suggestions to help here, but perhaps the authors could have a close read of this part to see if it may be improved.

Then the procedure is applied and the CALIOP data set presented over the 14 year period. The results are quite interesting and I appreciate the separation of the stratosphere into three layers associated with their relevance to cross tropopause transport.

[Figure]

This separation led to some nice generalizations concerning the volcanic eruptions which occurred over the 14 year period and how these eruptions influenced AOD and hence the cooling associated with stratospheric aerosol.

The paper is interesting, especially once the data are discussed, and should be published. There are a few things that need to be taken care of before that happens.

Aside from the detailed comments below the authors should add labels a), b), ... to all the figure panels. The authors are very good at explaining what each of the a), b), ... panels are in the captions, and refer to Figure Xa in the text. But none of the panels in the figures have labels.

Overall the writing is quite good, but there are a few awkward places. I have tried to flag these. One general comment the phrase "in order" is never necessary and can be deleted with no change in the sentence meaning.

1.9-11. Why switch between CALIOP and CALIPSO in the abstract when neither are defined. CALIOP is the instrument and should suffice in both places.

1.19-20. Awkward sentence, beginning with "of which". Try. Trends in the abundance of aerosol particles are an important component of the climate system, although their influence on climate is still highly uncertain. . .

2.10-14. Change forming to contributing. If OCS is known to form the Junge layer (which it isn't) then why the discussion about so2?

2.21 LMS? -

5.20 residing . . .

6.9 becomes . . .

7.29-31. This is confusing. Why would you expect aerosol scattering to compare with extinction? There is at least a factor of 50 between them. In Fig. 3 it is much more. The AS is noted in a range of 1e-8 to 1e-7, while the extinctions range from 0.02-0.1.

Fig. 3. There are no labels a), b), . . . Why is there a big 0 between the panels at the bottom?

8.1 Where is Fig. 2b? There are no labels on Figure 2 either. Why strangely?

11.2 first 8 months . . .

11.3. Are these the averages, maxima, value after 8 months, or?

11.28-29, "A feature appearing in the southern tropics in April has been identified as smoke from bush fires in February 2009 in Victoria, Australia (Vernier et al., 2011)." Is this the feature near 20 km, if so it is worth mentioning the altitude as this may be surprising for some. Best to be clear.

Figure 6. Perhaps the vertical solid lines represent the tropical eruptions? The figure caption is confusing. Please add short labels to the eruptions, which are listed in Table 1, so the reader can easily see the effects from Sarychev, Calbuco, and Kasatochi, without having to search them out. Similar labels would be appreciated in Figures, 5, 7-9.

13.18-19. What PV levels are considered the LMS. This is needed to understand the statement, "the aerosol signal is strong also in the lowest LMS layer of Fig. 6a". What is the lowest LMS layer?

13.25. The discussion previous to this conclusion has been descriptive of Figure 6, but it's not clear why the last half of this sentence holds. The volcanoes affected all the PV layers investigated more or less equally. It would be nice to see a PV level that wasn't affected, then maybe there would be a justification for the choice.

14.6-7. This conclusion about the source of the springtime increase in aerosol scattering is much too strong based on the evidence given. What do "These observations" refer to, to the work here or that by Martinsson? Perhaps they "suggest", but "indicate", no. Much more work would be required to come to such a definite conclusion.

[Figure]

Figure 8. Why don't the sum of the three lines add up to the total? There is usually at least a difference of 0.002 in AOD between the sum of the three layers and the total.

14.26-29. This is a bit surprising. Can some more detail be provided? Using Figure 6 the spring time maxima in 2007 and 2008 indicate SR in the lowest layers equivalent to the SR after Nabro, which shows a large AOD. Is it the fact that the impact of the dust layers, or whatever is causing this, so narrow that the impact on AOD is small?

15.3. The small peak near the equator in the upper layer can hardly be considered to be "high AOD". It is only slightly higher than in the mid latitudes.

15.26-31. It would be helpful to indicate here that Figure 7 is being discussed. This is the figure I used to follow the discussion.

16.4. "A small sudden increase in the AOD is observed in the southern extratropics approximately one and a half years after the Kelut eruption." At what level? Which figure is being referred to?

16.8-9. "The aerosol in the upper layer is eventually transported out to the next lower layer at midlatitudes, i.e the one located between the 380-470 K isentropes." It would be nice to know what figure is being used to make this claim. Figure 7 for example does not support this statement.

19.2. "... forcing, and is added ..."

---

## Referee Comment (RC2) · Anonymous Referee #2 · 23 Mar 2018

Review of Friberg et al. (2018)

Volcanic eruptions represent one of the largest source of natural variability of our climate system. Thus, the construction of long-term database of stratospheric aerosol optical depth is highly important to constrain global climate models. In this study, Friberg et al. (2018) used the CALIPSO space-born lidar to derive the time evolution of stratospheric AOD between 2005 and 2016. They proposed two new techniques to correct the effect of particle attenuation on retrieved optical parameters (backscatter and extinction) and remove Polar Stratospheric Clouds. After selecting a definition of the tropopause based on Potential Vorticity, they show time series of stratospheric AOD and discuss the influence of several volcanic eruptions. Overall, the paper is interesting, and the technique developed to correct the effect of particle attenuation is relatively

well explained. But there are several points which would need serious considerations before its publication in ACP.

1. Purpose of this work. The purpose of this study is not clear. Do the authors intend to create a new aerosol dataset from the CALIPSO data? If yes, do they have a plan for archiving the data and make them publicly available? If this is the case, is there any established collaboration with the CALIPSO team to work on this dataset or it is an independent endeavor?

2. Lack of validation. This study does not make use of additional datasets to compare/validate the retrieved AOD. Several satellite datasets (e.g. OSIRIS, OMPS) are available and could provide a source of validation data but are not used. What about In situ data from the CARIBIC program such as aerosol size distribution in the Lowermost stratosphere? Why not to use those data to infer lidar ratio using Mie Calculations?

3. Retrieving AOD from CALIPSO. The authors propose an approach to correct the particle attenuation effect which is especially important after a significant volcanic eruption. A major issue with the proposed technique is its dependency to the type of volcanic eruptions. The major assumption of the correction technique is to assume that the Upper Troposphere is clear of volcanic aerosol, but this is not always the case as shown after the Kasatochi eruption. Any techniques applied for this purpose should be independent from volcanoes and therefore could only be achieved by the iterative approach developed in Hostetler et al. (2006). The overall impact on the corrected AOD is relatively small (impact between 4-7 %). The authors never discussed thoughtfully the other sources of uncertainties that could have bigger influences (e.g. calibration of the lidar, lidar ratio conversion factor). For example, they rapidly mentioned that the lidar ratio values of 50sr used to convert backscatter into extinction agree with Prata et al. (2017). This is not correct, Prata et al. (2017) found a mean lidar ratio of 69 sr for the Cordon plume, 66 for Kasatochi and 63 for Sarychev. This would increase the volcanic AOD during volcanically influenced periods by 30-40 %. The lidar ratio assumption is therefore one of the main source of uncertainty for AOD retrieved from
CALIPSO but poorly discussed here.

4. English language. This is an overall issue which could be difficult to address without a native English speaker person. However, the level of English in the paper is relatively poor and would need to be improved. I recommend the co-authors of the paper to take part of this effort to improve the English.

---

## Author Comment (AC1) · 4 May 2018

We would like to start by thanking the reviewer for the many constructive comments that help us improve the manuscript. The reviewer's questions and remarks have been answered in the text below, where reviewer comments are marked in black and our responses in this blue color.
* * *
This contribution to our understanding of stratospheric aerosol optical depth (AOD) focuses on the CALIOP backscatter data set after it is cleared of clouds. A significant fraction of the paper is devoted to a procedure to correct CALIOP measurements beneath volcanic clouds when the laser is attenuated by the volcanic cloud. The procedure is relatively straight forward, although I found the discussion of how the threshold is set to instigate the procedure, and the relevance of Figure 3 to that procedure, confusing. I don't have specific suggestions to help here, but perhaps the authors could have a close read of this part to see if it may be improved.

Then the procedure is applied and the CALIOP data set presented over the 14 year period. The results are quite interesting and I appreciate the separation of the stratosphere into three layers associated with their relevance to cross tropopause transport. This separation led to some nice generalizations concerning the volcanic eruptions which occurred over the 14 year period and how these eruptions influenced AOD and hence the cooling associated with stratospheric aerosol.

The paper is interesting, especially once the data are discussed, and should be published. There are a few things that need to be taken care of before that happens.

Aside from the detailed comments below the authors should add labels a), b), . . . to all the figure panels. The authors are very good at explaining what each of the a), b), . . . panels are in the captions, and refer to Figure Xa in the text. But none of the panels in the figures have labels. We apologize for overlooking this, and will include the indexes to all subfigures.

Overall the writing is quite good, but there are a few awkward places. I have tried to flag these. One general comment the phrase "in order" is never necessary and can be deleted with no change in the sentence meaning. Thank you for pointing that out. We have included the suggested changes, and make further language corrections adjusting to comment #4 by the second reviewer.

1.9-11. Why switch between CALIOP and CALIPSO in the abstract when neither are defined. CALIOP is the instrument and should suffice in both places. We agree with the reviewer. We will use CALIOP throughout the manuscript, and add the definition in the abstract.

1.19-20. Awkward sentence, beginning with "of which". Try. Trends in the abundance of aerosol particles are an important component of the climate system, although their influence on climate is still highly uncertain. . . We have used the reviewer's suggestion.

2.10-14. Change forming to contributing. If OCS is known to form the Junge layer (which it isn't) then why the discussion about so2? The reviewer is right, and we have changed accordingly.

2.21 LMS? – We forgot to define the lowermost stratosphere. Thank you for pointing that out.

5.20 residing . . . We changed accordingly.

6.9 becomes . . . We changed accordingly.

7.29-31. This is confusing. Why would you expect aerosol scattering to compare with extinction? There is at least a factor of 50 between them. In Fig. 3 it is much more. The AS is noted in a range of 1e-8 to 1e-7, while the extinctions range from 0.02-0.1. The units are not of importance for the comparison. It is simply done to highlight that the Level 2 extinction data do not follow that of the volcanic aerosol. We understand that it is not obvious for the reader, and therefore added a sentence to clarify this.

Fig. 3. There are no labels a), b), . . . Why is there a big 0 between the panels at the bottom? We do not see this. Can it be that the reviewer refers to the initially submitted version of the manuscript (from December, 2017)? We added the subfigure indices between that version and the current (version submitted on January 25, 2018)

8.1 Where is Fig. 2b? There are no labels on Figure 2 either. Why strangely? We apologize for missing this. Subfigure indexing is added in the new version (May, 2018).

11.2 first 8 months . . . We changed this.

11.3. Are these the averages, maxima, value after 8 months, or? They are the average, and we have now added that info in the text.

11.28-29, "A feature appearing in the southern tropics in April has been identified as smoke from bush fires in February 2009 in Victoria, Australia (Vernier et al., 2011)." Is this the feature near 20 km, if so it is worth mentioning the altitude as this may be surprising for some. Best to be clear. It is the feature around 20 km altitude. We agree with the reviewer and have clarified this in the manuscript.

Figure 6. Perhaps the vertical solid lines represent the tropical eruptions? The figure caption is confusing. Please add short labels to the eruptions, which are listed in Table 1, so the reader can easily see the effects from Sarychev, Calbuco, and Kasatochi, without having to search them out. Similar labels would be appreciated in Figures, 5, 7-9. The indexing was incorrect. Thank you for pointing that out. We change this, and add indexing in the figures.

13.18-19. What PV levels are considered the LMS. This is needed to understand the statement, "the aerosol signal is strong also in the lowest LMS layer of Fig. 6a". What is the lowest LMS layer? All of the PV-levels in the figure pertain to the LMS. The lowest LMS layer is 1.5-2 PVU. We will clarify this in the manuscript.

13.25. The discussion previous to this conclusion has been descriptive of Figure 6, but it's not clear why the last half of this sentence holds. The volcanoes affected all the PV layers investigated more or less equally. It would be nice to see a PV level that wasn't affected, then maybe there would be a justification for the choice.

The value of 1.5 PVU is the lowest dynamic tropopause commonly used. We find volcanism to impact all the way down to this lowest (stratospheric) PV-level (and in the case of Kasatochi there is an impact also below this PV-level). Therefore, we choose the 1.5 PVU to represent the lower altitude limit where volcanism impacts the stratospheric aerosol. This will be clarified in the manuscript.

14.6-7. This conclusion about the source of the springtime increase in aerosol scattering is much too strong based on the evidence given. What do "These observations" refer to, to the work here or that by Martinsson? Perhaps they "suggest", but "indicate", no. Much more work would be required to come to such a definite conclusion. We agree with the reviewer and have changed to "suggest".

Figure 8. Why don't the sum of the three lines add up to the total? There is usually at least a difference of 0.002 in AOD between the sum of the three layers and the total. We have controlled this and do not find any discrepancy, neither in the codes out-print, nor in the figure in the manuscript. We find that the three lines (representing the three layers) in each subfigure adds up to the total (red lines). We would be grateful if the reviewer could point out specific coordinates as an example of the stated discrepancy.

14.26-29. This is a bit surprising. Can some more detail be provided? Using Figure 6 the spring time maxima in 2007 and 2008 indicate SR in the lowest layers equivalent to the SR after Nabro, which shows a large AOD. Is it the fact that the impact of the dust layers, or whatever is causing this, so narrow that the impact on AOD is small?

The seasonal increase in AOD from dust (and other tropospheric sources), are estimated to constitute *less than 5%* of that of the AOD increase from Nabro. The dust signals are rapidly decreasing in strength in the first 2 km above the tropopause (1.5-5.5 PVU, the ExTL). Thus, they constitute a small fraction of the total stratospheric AOD, since they are contained within such a small fraction of the stratospheric air-mass. The picture is the opposite for the Nabro aerosol. The sulfate concentrations are increasing from the tropopause into the stratosphere, and this effect is larger for tropical eruptions than for the local eruptions in midlatitudes. In midlatitudes the subsidence of air (Nabro's aerosol) from the 380 K isentrope through the LMS to the tropopause takes several months, during which the stratospheric air gradually mixes with tropospheric air which decreases the sulfate concentrations and induces a gradient of sulfate from the tropopause throughout the LMS. The sulfur gradient and the weak Nabro aerosol signals around the tropopause are both evident in Figure 5a. We have estimated that *the Nabro aerosol in the ExTL constituted less than 6%* of the total stratospheric AOD during the period of 1-8 months after its eruption.

This may be counterintuitive from the appearance of figure 6. We will therefore add discussion on this in section where the figure is presented.

15.3. The small peak near the equator in the upper layer can hardly be considered to be "high AOD". It is only slightly higher than in the mid latitudes. The reviewer is right. We have reformulated our statement.

15.26-31. It would be helpful to indicate here that Figure 7 is being discussed. This is the figure I used to follow the discussion. Thanks for pointing that out. We have added a reference to Figure 7.

16.4. "A small sudden increase in the AOD is observed in the southern extratropics approximately one and a half years after the Kelut eruption." At what level? Which figure is being referred to? This is indeed confusing. We have added a more descriptive text.

16.8-9. "The aerosol in the upper layer is eventually transported out to the next lower layer at midlatitudes, i.e the one located between the 380-470 K isentropes." It would be nice to know what figure is being used to make this claim. Figure 7 for example does not support this statement.

The AOD in Figure 7b indicates this. For example, the AOD in the northern midlatitudes of the layer 380-470 K was higher in year 2007 and beginning of 2008 compared to the background (year 2013). The increased AOD in midlatitudes most likely comes from subsidence from the overlying layer. We will make changes to the text to clarify this.

19.2. ". . . forcing, and is added . . ." We will make the suggested change.

---

## Author Comment (AC2) · 4 May 2018

We would like to start by thanking the reviewer for helping us improve the manuscript. The reviewer's questions and remarks have been answered in the text below, where reviewer comments are marked in black and our responses in this blue color.
* * *
Volcanic eruptions represent one of the largest source of natural variability of our climate system. Thus, the construction of long-term database of stratospheric aerosol optical depth is highly important to constrain global climate models. In this study, Friberg et al. (2018) used the CALIPSO space-born lidar to derive the time evolution of stratospheric AOD between 2005 and 2016. They proposed two new techniques to correct the effect of particle attenuation on retrieved optical parameters (backscatter and extinction) and remove Polar Stratospheric Clouds. After selecting a definition of the tropopause based on Potential Vorticity, they show time series of stratospheric AOD and discuss the influence of several volcanic eruptions. Overall, the paper is interesting, and the technique developed to correct the effect of particle attenuation is relatively well explained. But there are several points which would need serious considerations before its publication in ACP.

1. Purpose of this work. The purpose of this study is not clear. Do the authors intend to create a new aerosol dataset from the CALIPSO data? If yes, do they have a plan for archiving the data and make them publicly available? If this is the case, is there any established collaboration with the CALIPSO team to work on this dataset or it is an independent endeavor?

The purpose of this work is to investigate the stratospheric aerosol in a period when several volcanic eruptions affected the stratosphere. Compared to previous studies we have put emphasis on understanding the aerosol load in relation to important transport features of the stratosphere. We have also included a larger part of the stratosphere compared with previous studies, approaching a complete coverage of the stratospheric aerosol of the period. We highly appreciate the work of the CALIPSO team, and would be interested in collaboration just as we have been in the past (Andersson et al., 2015).

2. Lack of validation. This study does not make use of additional datasets to compare/ validate the retrieved AOD. Several satellite datasets (e.g. OSIRIS, OMPS) are available and could provide a source of validation data but are not used. What about In situ data from the CARIBIC program such as aerosol size distribution in the Lowermost stratosphere? Why not to use those data to infer lidar ratio using Mie Calculations?

CALIOP data are validated already in for example publications by Vernier, and used in the Science paper of Solomon et al. (2011). Our methods to retrieve the AOD are not significantly different from Vernier's and that used in our previous CALIOP publication in Nature Communications 2015 (Andersson et al., 2015). We have also validated that we retrieve the same scattering and AOD values (at 20-30 km) as in Vernier et al. (2011). We will compare our AODs with those in Rieger et al. (2015), Vernier et al. (2011), and Thomason et al. (2018), and add a more thorough description about the lidar ratio in the manuscript.

Our intension is to discuss the volcanic impact from the global observations of CALIOP, and to include the entire stratosphere. Furthermore, a paper on comparison between CALIOP and CARIBIC is in preparation by one of the co-authors.

Our paper extends the study by Andersson et al. (2015) highlighting the significance of the LMS, which has been neglected until very recently, and the overlying stratosphere has to our knowledge not been separated in individual layers before.

*3. Retrieving AOD from CALIPSO. The authors propose an approach to correct the particle attenuation effect which is especially important after a significant volcanic eruption. A major issue with the proposed technique is its dependency to the type of volcanic eruptions. The major assumption of the correction technique is to assume that the Upper Troposphere is clear of volcanic aerosol, but this is not always the case as shown after the Kasatochi eruption. Any techniques applied for this purpose should be independent from volcanoes and therefore could only be achieved by the iterative approach developed in Hostetler et al. (2006). The overall impact on the corrected AOD is relatively small (impact between 4-7 %).*

It is true that it depends on the type of eruption and cannot be used directly for all types of eruptions. Kasatochi is a special case. In-situ observations by CARIBIC shows that volcanic elevations of the stratospheric aerosol load generally have small impact on the UT aerosol concentrations. Thus, the proposed method can be used directly for eruptions which injections mainly reached above the UT, and by our method we risk doing a small underestimation of the true volcanic elevations.

We believe that the method described in Hostetler et al. (2006)/Young et al. (2005) is good for correcting for attenuation in detected volcanic layers, but that it is evident that the method cannot detect the volcanic aerosol once it has mixed with the background. Hence, the two methods have different weaknesses.

The computed changes are relatively small in perspective to the total stratospheric AOD for the period studied here, but could become much more important for stronger eruptions. The attenuation depends on the overlying aerosol column, and is thus particularly important for the representation of the lowest layer of the stratosphere, i.e. the LMS.

*The authors never discussed thoughtfully the other sources of uncertainties that could have bigger influences (e.g. calibration of the lidar, lidar ratio conversion factor).* We agree that these uncertainties are important and that the attenuation corrections made in the manuscript are small in comparison to e.g. that in the lidar ratio, and we decided to include a description in the methods section according to the suggestion of the reviewer. Our corrections make a significant difference for the retrieved aerosol scattering, especially in the LMS.

*For example, they rapidly mentioned that the lidar ratio values of 50sr used to convert backscatter into extinction agree with Prata et al. (2017). This is not correct, Prata et al. (2017) found a mean lidar ratio of 69 sr for the Cordon plume, 66 for Kasatochi and 63 for Sarychev. This would increase the volcanic AOD during volcanically influenced periods by 30-40 %. The lidar ratio assumption is therefore one of the main source of uncertainty for AOD retrieved from CALIPSO but poorly discussed here.*

We agree with the reviewer that the motivation for using the factor 50 sr was discussed too shortly.

The assumption of value on the lidar ratio is indeed a large uncertainty for the retrievals by CALIOP. The figures by Prata et al. (2017) are higher than those used in our manuscript, but according to the uncertainties reported in Prata et al. (2017) the factor 50 sr is not significantly different from the values reported in their paper, but we are on the lower side of their estimations. They reported *means and standard deviations* for the lidar ratios of volcanic layers to be *69±13 sr* (Puyehue-Cordón Caulle, Jun 2011), *66±19 sr* (Kasatochi, Aug 2008) and *63±14 sr* (Sarychev).

There are differences in methodology in our manuscript and that of the Prata et al. (2017) paper. They found relatively fresh volcanic aerosol plumes in the stratosphere and computed the AOD from that, while we use zonal means, i.e. mixtures of layers of volcanic aerosol and the stratospheric background aerosol. Also, the values reported in Prata et al. 2017 are based on fresh aerosol. Aging of the aerosol and mixing with background changes the particle size distribution. Thus, the lidar ratio is not necessarily equal for the fresh volcanic aerosol, and aerosol that has been aged and mixed with the background. Most studies that dealt with stratospheric aerosol show lower lidar ratios than those reported in Prata et al., 2017. In a most recent study Thomason et al. (2018) find the CALIOP lidar ratio to have a value of 53 sr. We therefore believe that using the values reported in Jäger and Deshler (2003) is well-suited for our study.

We have decided to include a more thorough discussion on the uncertainties in the lidar ratio. We will also point out (in the manuscript) that we used a conservative value of the lidar ratio, along with the comparison to other AOD data-sets (Rieger et al. (2015), Vernier et al. (2011), and Thomason et al. (2018)).

*4. English language. This is an overall issue which could be difficult to address without a native English speaker person. However, the level of English in the paper is relatively poor and would need to be improved. I recommend the co-authors of the paper to take part of this effort to improve the English.*

We are aware that we as non-native speakers are in a difficult position language-wise. Reviewer #1 wrote that "…Overall the *writing is quite good, but there are a few awkward places*…", and also provided a thorough list of suggested changes. We have corrected many language-mistakes by taking into account the many points made by Reviewer #1, and gone through the text with a critical eye. We believe that these changes, as well as all other changes, have improved the manuscript.

**References**

Andersson, S. M., Martinsson, B. G., Vernier, J. P., Friberg, J., Brenninkmeijer, C. A. M., Hermann, M., Van Velthoven, P. F. J. and Zahn, A.: Significant radiative impact of volcanic aerosol in the lowermost stratosphere, Nat. Commun., 6(May), 1–8, doi:10.1038/ncomms8692, 2015.

Hostetler, C. a, Liu, Z., Reagan, J., Vaughan, M., Winker, D., Osborn, M., Hunt, W. H., Powell, K. a and Trepte, C.: CALIOP Algorithm Theoretical Basis Document - Calibration and Level 1 Data Products, , (PC-SCI-201 Release 1.0) [online] Available from: http://www-calipso.larc.nasa.gov/resources/pdfs/PC-SCI-201v1.0.pdf, 2006.

Jäger, H. and Deshler, T.: Erratum: Lidar backscatter to extinction, mass and area conversions for stratospheric aerosols based on midlatitude balloonborne size distribution measurements (Geophysical Research Letters (2002) 29:19 (1929) DOI:10.1029/2002GL015609), Geophys. Res. Lett., 30(7), 1–4, doi:10.1029/2003GL017189, 2003.

Prata, A. T., Young, S. A., Siems, S. T. and Manton, M. J.: Lidar ratios of stratospheric volcanic ash and sulfate aerosols retrieved from CALIOP measurements, Atmos. Chem. Phys., 17(13), 8599–8618, doi:10.5194/acp-17-8599-2017, 2017.

Rieger, L. A., Bourassa, A. E. and Degenstein, D. A.: Merging the OSIRIS and SAGE II stratospheric aerosol records, J. Geophys. Res., 120(17), 8890–8904, doi:10.1002/2015JD023133, 2015.

Solomon, S., Daniel, J. S., Neely, R. R., Vernier, J.-P., Dutton, E. G. and Thomason, L. W.: The Persistently Variable "Background" Stratospheric Aerosol Layer and Global Climate Change, Science (80-. )., 333(6044), 866–870, doi:10.1126/science.1206027, 2011.

Thomason, L. W., Ernest, N., Millán, L., Rieger, L., Bourassa, A., Vernier, J. P., Manney, G., Luo, B., Arfeuille, F. and Peter, T.: A global space-based stratospheric aerosol climatology: 1979-2016, Earth Syst. Sci. Data, 10(1), 469–492, doi:10.5194/essd-10-469-2018, 2018.

Vernier, J. P., Thomason, L. W., Pommereau, J. P., Bourassa, A., Pelon, J., Garnier, A., Hauchecorne, A., Blanot, L., Trepte, C., Degenstein, D. and Vargas, F.: Major influence of tropical volcanic eruptions on the stratospheric aerosol layer during the last decade, Geophys. Res. Lett., 38(12), 1–8, doi:10.1029/2011GL047563, 2011.

Young, S. A., Winker, D. M., Noel, V., Vaughan, M. A., Hu, Y. and Kuehn, R. E.: Cloud-Aerosol Lidar Infrared Pathfinder Satellite Observations CALIOP Algorithm Theoretical Basis Document Document No: PC-SCI-201., 2005.